# DynaVol: Unsupervised Learning for Dynamic Scenes through Object-Centric Voxelization

**Yanpeng Zhao**[*]     **Siyu Gao**[*]     **Yunbo Wang** [†]     **Xiaokang Yang**
MoE Key Lab of Artificial Intelligence, AI Institute, Shanghai Jiao Tong University
`{zhao-yan-peng, siyu.gao, yunbow, xkyang}@sjtu.edu.cn`
`https://sites.google.com/view/dynavol/`

## Abstract

Unsupervised learning of object-centric representations in dynamic visual scenes is challenging. Unlike most previous approaches that learn to decompose 2D images, we present DynaVol, a 3D scene generative model that unifies geometric structures and object-centric learning in a differentiable volume rendering framework. The key idea is to perform *object-centric voxelization* to capture the 3D nature of the scene, which infers the probability distribution over objects at individual spatial locations. These voxel features evolve through a canonical-space deformation function, forming the basis for global representation learning via slot attention. The voxel and global features are complementary and leveraged by a compositional NeRF decoder for volume rendering. DynaVol remarkably outperforms existing approaches for unsupervised dynamic scene decomposition. Once trained, the explicitly meaningful voxel features enable additional capabilities that 2D scene decomposition methods cannot achieve: it is possible to freely edit the geometric shapes or manipulate the motion trajectories of the objects.

## 1 Introduction

Unsupervised learning of the physical world is of great importance but challenging due to the intricate entanglement between the spatial and temporal information (Wu et al., 2015; Santoro et al., 2017; Greff et al., 2020). Existing approaches primarily leverage the consistency of the dynamic information across consecutive video frames but tend to ignore the 3D nature, resulting in a multi-view mismatch of 2D object segmentation (Kabra et al., 2021; Elsayed et al., 2022; Singh et al., 2022). In this paper, we explore a novel research problem of unsupervised 3D dynamic scene decomposition. Different from the previous 2D counterparts, our method naturally ensures 3D-consistent scene understanding and can motivate downstream tasks such as scene editing. We believe that an effective object-centric representation learning method should satisfy two conditions: First, it should capture the *time-varying local structures* of the visual scene in a 3D-consistent way, which requires the ability to decouple the underlying dynamics of each object from visual appearance; Second, it should obtain a *global understanding* of each object, which is crucial for downstream tasks such as relational reasoning.

Accordingly, we propose to learn two sets of object-centric representations: one that represents the local spatial structures using time-aware voxel grids, and another that represents the time-invariant global features of each object. To achieve this, we introduce **DynaVol**, which incorporates **object-centric voxelization** into the unsupervised learning framework of inverse rendering. The key idea is that object-centric voxelization allows us to *infer the probability distribution over objects at individual spatial locations*, thereby naturally facilitating 3D-consistent scene decomposition (see Figure 1).

DynaVol consists of three network components in the test phase: (i) A dynamics module that learns the transitions of voxel grid features in canonical space over time; (ii) A volume slot attention module that progressively refines the object-level, time-invariant global features by aggregating the voxel representations; (iii) An object-centric, compositional neural radiance field (NeRF) for view synthesis, driven by both the local and global object representations. In contrast to prior research that focuses on

---

[*]Equal contribution.
[†]Corresponding author: Yunbo Wang.

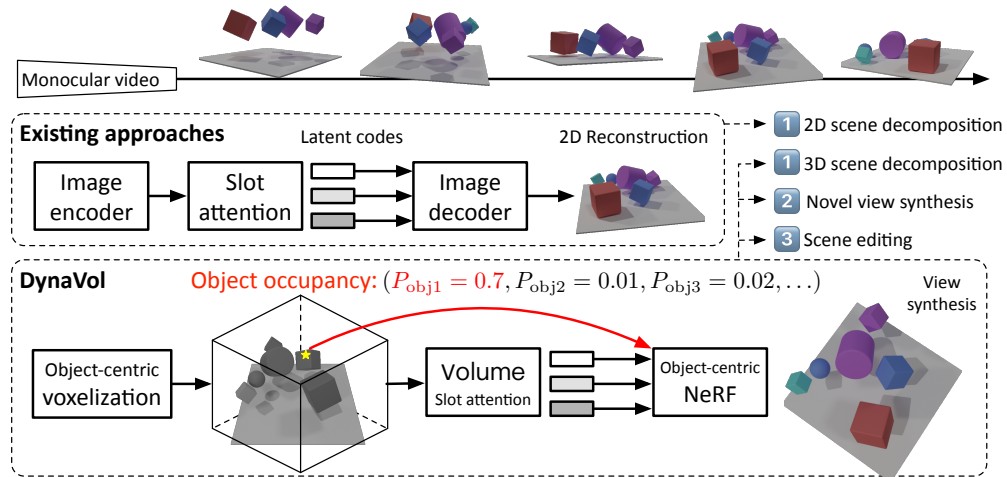

Figure 1: DynaVol explores an unsupervised object-centric voxelization approach for dynamic scene decomposition. Unlike its 2D counterparts, such as SAVi (Kipf et al., 2022), DynaVol ensures 3D consistency and provides additional capabilities, *e.g.*, novel view synthesis and scene editing.

decomposing 2D images (Kipf et al., 2022; Sajjadi et al., 2022; Elsayed et al., 2022), our unsupervised voxelization approach provides two additional advantages beyond novel view synthesis. First, it allows for fine-grained separation of object-centric information in 3D space without any geometric priors. Second, it enables direct scene editing (*e.g.*, object removal, replacement, and trajectory modification) that is not feasible in existing video decomposition methods. This is done by directly manipulating the voxel grids or the learned deformation function without the need for further training.

In our experiments, we initially evaluate DynaVol on simulated 3D dynamic scenes that contain different numbers of objects, diverse motions, shapes (such as cubes, spheres, and real-world shapes), and materials (such as rubber and metal). On the simulated dataset, we can directly assess the performance of DynaVol for scene decomposition by projecting the object-centric volumetric representations onto 2D planes, and compare it with existing approaches, such as SAVi (Kipf et al., 2022) and uORF (Yu et al., 2022). Additionally, we demonstrate the effectiveness of DynaVol in novel view synthesis and dynamic scene editing using real-world videos.

## 2 PROBLEM SETUP

We assume a sparse set of visual observations of a dynamic scene $\{\mathbf{I}_{t=1}^v, \mathbf{T}_{t=1}^v\}_{v=1}^V$ at the initial timestamp and a set of RGB images $\{\mathbf{I}_t, \mathbf{T}_t\}_{t=2}^T$ collected with a moving monocular camera, where $\mathbf{I}_t^v \in \mathbb{R}^{H \times W \times 3}$ are images acquired under camera poses $\mathbf{T}_t^v \in \mathbb{R}^{4 \times 4}$. $T$ is the length of video frames and $V$ is the number of views at the initial timestamp. The goal is to decompose each object in the scene by harnessing the underlying space-time structures present in the visual data without additional information. Please note that we also consider scenarios where (i) only one camera pose is available at the first timestamp (*i.e.*, $V = 1$) and (ii) images are captured by a monocular camera along a smooth moving trajectory. These configurations can be easily achieved in real-world scenes.

## 3 METHOD

In this section, we first discuss the overall framework of DynaVol. Subsequently, we introduce the concept of object-centric voxel representations and provide the details of each network component. Finally, we present the three-stage training procedure of our approach.

### 3.1 OVERVIEW OF DYNAVOL

DynaVol is trained in an inverse graphics framework to synthesize $\{\mathbf{I}_{t=1}^v\}_{v=1}^V$ and $\{\mathbf{I}_t\}_{t=2}^T$ without any further supervision. Formally, the goal is to learn an object-centric projection of $(\mathbf{x}, \mathbf{d}, t) \to \{(c_n, \sigma_n)\}_{n=1}^N$, where $\mathbf{x} = (x, y, z)$ is a 3D point sampled by the neural renderer and $N$ is the predefined number of slots, which is assumed to be larger than the number of objects. The core of our approach is to introduce an object-centric 4D voxel representation, denoted by $\mathcal{V}_t \in \mathbb{R}^{N \times N_x \times N_y \times N_z}$, which maintains the time-vary density $\{\sigma_n\}_{n=1}^N$ of each possible object. The renderer estimates the density and color for each object at view direction $\mathbf{d}$ and re-combines $\{(c_n, \sigma_n)\}_{n=1}^N$ to approach the target pixel value. As shown in Figure 2, DynaVol consists of three network components: (i) The

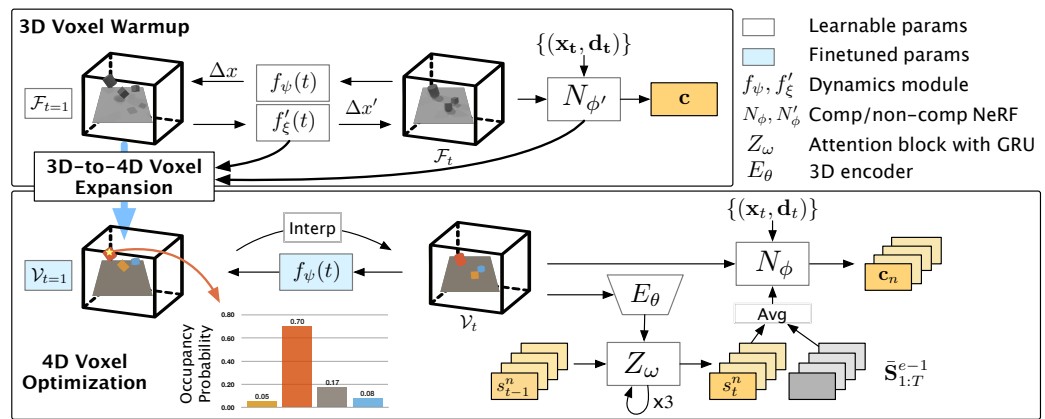

Figure 2: The architecture of DynaVol consists of three groups of network components: the bi-directional dynamics modules $(f_\psi, f'_\xi)$, the slot attention module with a GRU and a 3D volume encoder $(Z_\omega, E_\theta)$, and neural rendering modules $(N_{\phi'}, N_\phi)$ based on 3D and 4D voxels respectively. The training scheme involves a 3D voxel warmup stage (top), a 3D-to-4D voxel expansion stage (middle, whose pseudocode is given in Appendix.B), and a 4D voxel optimization stage (bottom).

bi-directional deformation networks $f_\psi$ and $f'_\xi$ that learn the canonical-space transitions over time of the object-centric voxel representations contained within $\mathcal{V}_t$; (ii) The volume encoder $E_\theta$ and a slot attention block $Z_\omega$ that work collaboratively to refine a set of global slot features $\mathbf{S}_t = \{s^n_t\}^N_{n=1}$ iteratively, which convey time-invariant object information; (iii) The image renderers, which include a compositional NeRF denoted by $N_\phi$ that jointly uses $\mathcal{V}_t$ and $\mathbf{S}_t$ to generate the observed images, and a non-compositional NeRF denoted by $N_{\phi'}$ that is only used to initialize the voxel representations.

DynaVol involves three stages in the training phase: First, a 3D voxel warmup stage that learns to obtain a preliminary understanding of the geometric and dynamic priors. Second, a 3D-to-4D voxel expansion stage that extends the 3D density grids $\mathcal{F}_{t=1} \in \mathbb{R}^{N_x \times N_y \times N_z}$ to 4D dimension, initializing $\mathcal{V}_{t=1}$ using the connected components algorithm. Third, a 4D voxel optimization stage that optimizes $\{\mathcal{V}_{t=1}, \mathbf{S}_{t=0}, f_\psi, E_\theta, Z_\omega, N_\phi\}$ to refine the object-centric voxel representations.

### 3.2 OBJECT-CENTRIC VOXEL REPRESENTATIONS

We extend the idea of using 3D voxel grids to maintain the volume density for neural rendering with 4D voxel grids, denoted as $\mathcal{V}_t$. The additional dimension indicates the occupancy probabilities of each object within each grid cell. The occupancy probability $\{\sigma_n\}^N_{n=1}$ at an arbitrary 3D location can be efficiently queried through the trilinear interpolation sampling method:

$$\text{Interp}(\mathbf{x}, \mathcal{V}_t) : (\mathbb{R}^3, \mathbb{R}^{N \times N_x \times N_y \times N_z}) \rightarrow \mathbb{R}^N, \tag{1}$$

where $(N_x, N_y, N_z)$ are the resolutions of $\mathcal{V}_t$. To achieve sharp decision boundaries during training, we apply the Softplus activation function to the output of trilinear interpolation.

### 3.3 MODEL COMPONENTS

**Canonical-space dynamics modeling.** As shown in Figure 2, we use a dynamics module $f_\psi$ to learn the deformation field from $\mathcal{V}_{t=1}$ at the initial timestamp to its canonical space variations over time. Notably, in the warmup stage, $f_\psi$ learns to transform the density values in $\mathcal{F}_t$. Given a 3D point $\forall \mathbf{x}_i \in \{\mathbf{x}\}$ at an arbitrary time, $f_\psi(\mathbf{x}_i, t)$ predicts a position movement $\Delta \mathbf{x}_i$, so that we can transform $\mathbf{x}_i$ to the scene position at the first moment by $\mathbf{x}_i + \Delta \mathbf{x}_i$. We then query the occupancy probability from $\mathcal{V}_{t=1}$ by $\widehat{\mathcal{V}}_t = \text{Interp}\left((\mathbf{x}_i + f_\psi(\mathbf{x}_i, t)), \mathcal{V}_{t=1}\right)$. Notably, we encode $\mathbf{x}_i$ and $t$ into higher dimensions via positional embedding. Additionally, in the warmup stage, we use another dynamics module $f'_\xi$ to model the forward movement $\Delta x'$ from initial movement to timestamp $t$. This module enables the calculation of a cycle-consistency loss, enhancing the coherence of the learned canonical-space transitions. Furthermore, it provides useful input features representing the forward dynamics starting from the initial time step for the subsequent connect components algorithm, which can improve the initialization of $\mathcal{V}_{t=1}$.

**Volume slot attention.** To progressively derive a set of global *time-invariant* object-level representations from the local time-varying voxel grid features, we employ volume slot attention in our approach. Specifically, we use a set of latent codes referred to as "slots" to represent these object-level features. This terminology is in line with prior literature on 2D static scene decomposition (Locatello

et al., 2020). The slots are randomly initialized from a normal distribution and progressively refined episode by episode throughout our second training stage. The *episode* refers to a training period that iterates from the initial timestamp to the end of the sequence. We denote the collection of slot features by $\mathbf{S}_t^e \in \mathbb{R}^{N \times D}$, where $e$ is the index of the training episode and $D$ is the feature dimensionality. At the beginning of each episode, we initialize the slot features as $\mathbf{S}_{t=0}^e = \bar{\mathbf{S}}_{1:T}^{e-1} = \frac{1}{T}\sum_{t=1}^T (\mathbf{S}_t^{e-1})$, where $\bar{\mathbf{S}}_{1:T}^{e-1}$ is the average of slot features across all timestamps in the previous episode. Each slot feature captures the time-invariant properties such as the appearance of each object, which enables the manipulation of the scene's content and relationships between objects. To bind the voxel grid representations to the corresponding object, at timestamp $t$, we pass $\mathcal{V}_t$ through a 3D CNN encoder $E_\theta$, which consists of 3 convolutional layers with ReLU. It outputs $N$ flattened features $\mathbf{h}_t \in \mathbb{R}^{M \times D}$, where $M$ represents the size of the voxel grids that have been reduced in dimensionality by the encoder. From an optimization perspective, a set of well-decoupled global slot features can benefit the separation of the object-centric volumetric representations. To refine the slot features, we employ the iterative attention block denoted by $Z_w$ to incorporate the flattened local features $\mathbf{h}_t$. In a single round of slot attention at timestamp $t \geq 1$, we have:

$$\mathcal{A}_t = \mathrm{softmax}_N\left(\frac{1}{\sqrt{D}}k(\mathbf{h}_t) \cdot q(\mathbf{S}_{t-1})^T\right), \quad W_t^{i,j} = \frac{\mathcal{A}_t^{i,j}}{\sum_{l=1}^M \mathcal{A}_t^{l,j}}, \quad \mathcal{U}_t = W^T \cdot v(\mathbf{h}_t), \quad (2)$$

where $\mathbf{S}_{t-1}$ denotes $\mathbf{S}_{t-1}^e$ in short, $(q, k, v)$ are learnable linear projections $\mathbb{R}^{D \to D}$ (Luong et al., 2015), such that $\mathcal{A}_t \in \mathbb{R}^{M \times N}$ and $\mathcal{U}_t \in \mathbb{R}^{N \times D}$, and $\sqrt{D}$ is a fixed softmax temperature (Vaswani et al., 2017). The resulted slots features are then updated by a GRU as $\mathrm{GRU}(\mathcal{U}_t, \mathbf{S}_{t-1})$. We update the slot features and obtain $\mathbf{S}_t$ by repeating the attention computation 3 times at each timestamp. For more in-depth analyses on volume slot attention, please refer to Appendix F.

**Object-centric renderer.** Previous compositional NeRF, like in uORF (Yu et al., 2022), typically uses an MLP to learn a continuous mapping from sampling point $\mathbf{x}$, viewing direction $\mathbf{d}$, and slot features to the emitted densities $\{\sigma_n\}$ and colors $\{c_n\}$ of different slots. Our neural renderer takes as inputs $\{\hat{s}_t^n\}_{n=1}^N = \mathrm{mean}(\mathbf{S}_t, \bar{\mathbf{S}}_{1:T}^{e-1})$ at timestamp $t$. As discussed above, $\bar{\mathbf{S}}_{1:T}^{e-1}$ is the averaged slot features in the previous episode, which can be more stable than the frequently refined features $\mathbf{S}_t$. We perform object-centric projections using an MLP: $N_\phi : (\mathbf{x}, \mathbf{d}, \{\hat{s}_t^n\}) \to \{c_n\}$, and query $\{\sigma_n\}$ directly from the voxel grids $\widehat{\mathcal{V}}_t$ at the corresponding timestamp. We use density-weighted mean to compose the predictions of $c_n$ and $\sigma_n$ for different objects, such that:

$$w_n = \sigma_n / \sum_{n=1}^N \sigma_n, \quad \overline{\sigma} = \sum_{n=1}^N w_n \sigma_n, \quad \overline{\mathbf{c}} = \sum_{n=1}^N w_n c_n, \quad (3)$$

where $\overline{\sigma}$ and $\overline{\mathbf{c}}$ is the output density and the color of a sampling point. We estimate the color $C(\mathbf{r})$ of a sampling ray with the quadrature rule (Max, 1995): $\widehat{C}(\mathbf{r}) = \sum_{i=1}^P T_i (1 - \exp(-\overline{\sigma}_i \delta_i)) \overline{\mathbf{c}}_i$, where $T_i = \exp(-\sum_{j=1}^{i-1} \overline{\sigma}_j \delta_j)$, $P$ is the number of sampling points in a certain ray, and $\delta_i$ is the distance between adjacent samples along the ray.

## 3.4 TRAINING

We train the entire model of DynaVol using neural rendering objective functions. At a specific timestamp, we take the rendering loss $\mathcal{L}_{\mathrm{Render}}$ between the predicted and observed pixel colors, the background entropy loss $\mathcal{L}_{\mathrm{Ent}}$, and the per-point RGB loss $\mathcal{L}_{\mathrm{Point}}$ following DVGO (Sun et al., 2022) as basic objective terms. $\mathcal{L}_{\mathrm{Ent}}$ can be viewed as a regularization to encourage the renderer to concentrate on either foreground or background. To enhance dynamics learning in the warmup stage, we design a novel cycle loss between $f_\psi$ and $f_\xi'$:

$$\mathcal{L}_{\mathrm{Render}} = \frac{1}{|\mathcal{R}|}\sum_{r \in \mathcal{R}}\left\|\widehat{C}(\mathbf{r}) - C(\mathbf{r})\right\|_2^2, \quad \mathcal{L}_{\mathrm{Ent}} = \frac{1}{|\mathcal{R}|}\sum_{r \in \mathcal{R}} -\widehat{w}_l^r \log(\widehat{w}_l^r) - (1 - \widehat{w}_l^r)\log(1 - \widehat{w}_l^r),$$

$$\mathcal{L}_{\mathrm{Point}} = \frac{1}{|\mathcal{R}|}\sum_{r \in \mathcal{R}}\left(\frac{1}{P_r}\sum_{i=0}^{P_r}\|\overline{\mathbf{c}}_i - C(\mathbf{r})\|_2^2\right), \quad \mathcal{L}_{\mathrm{Cyc}} = \frac{1}{|\mathcal{R}|}\sum_{r \in \mathcal{R}}\left(\frac{1}{P_r}\sum_{i=0}^{P_r}\left\|f_\psi(x_i, t) + f_\xi'(x_i', t)\right\|_2^2\right),$$

$$(4)$$

where $x_i' = x_i + f_\psi(x_i, t), i \in [0, P_r]$, $\mathcal{R}$ is the set of sampled rays in a batch, $P_r$ is the number of sampling points along ray $r$, and $\widehat{w}_l^r$ is the color contribution of the last sampling point obtained by $\widehat{w}_l^r = T_{P_r}(1 - \exp(-\sigma_{P_r}\delta_{P_r}))$. We discuss the three stages in the training phase below.

**3D voxel warmup stage.** To reduce the difficulty of learning the object-centric 4D occupancy grids, we optimize the 3D density grids $\mathcal{F}_{t=1} \in \mathbb{R}^{N_x \times N_y \times N_z}$ and warmup $f_\psi$ using $\{\mathbf{I}_{t=1}^v\}_{v=1}^V$ and $\{\mathbf{I}_t\}_{t=2}^T$. To enhance the coherence of the learned canonical-space dynamics, we train an additional module $f'_\xi$ to capture the forward deformation field $\Delta x'_{1 \to t}$ using $\mathcal{L}_{\text{Cyc}}$ in Eq. (4). $\Delta x'_{1 \to t}$ is used in the subsequent connect components algorithm to improve the initialization of $\mathcal{V}_{t=1}$. We train the bi-directional $(f'_\xi, f_\psi)$ and the non-compositional neural renderer $N_{\phi'}$ based on $\mathcal{F}_{t=1}$. The overall objective is defined as $\mathcal{L}_{\text{Warm}} = \sum_{t=1}^T (\mathcal{L}_{\text{Render}} + \alpha_p \mathcal{L}_{\text{Point}} + \alpha_e \mathcal{L}_{\text{Ent}} + \alpha_c \mathcal{L}_{\text{Cyc}})$. The hyperparameter values are adopted from prior literature (Liu et al., 2022).

**3D-to-4D voxel expansion stage.** We extend the 3D voxel grids $\mathcal{F}_{t=1}$ to 4D voxel grids $\mathcal{V}_{t=1}$ using the connected components algorithm, in which the expanded dimension corresponds to the number of slots $(N)$. The input features for the connected components algorithm involve the forward canonical-space transitions generated by $f'_\xi$ and the emitted colors by $N_{\phi'}$. The output clusters symbolize different objects, based on our assumption that voxels of the same object have similar motion and appearance features. More details are given in Appendix.B.

**4D voxel optimization stage.** In this stage, we finetune $f_\psi$ and $\mathcal{V}_{t=1}$ obtained in the first two stages respectively. Following an end-to-end training scheme, the dynamics module, volume slot attention mechanism, and compositional renderer collaboratively contribute to refining the object-centric voxel grids. The loss function in this stage is defined as $\mathcal{L}_{\text{Dyn}} = \sum_{t=1}^T (\mathcal{L}_{\text{Render}} + \alpha_p \mathcal{L}_{\text{Point}} + \alpha_e \mathcal{L}_{\text{Ent}})$, where we finetune $(\mathcal{V}_{t_0}, \psi)$ and train $(\theta, \omega, \phi)$ from the scratch.

## 4 EXPERIMENTS

### 4.1 EXPERIMENTAL SETUP

**Datasets.** We build the 8 synthetic dynamic scenes in Table 1 using the Kubric simulator (Greff et al., 2022). Each scene spans 60 timestamps and contains different numbers of objects in various colors, shapes, and textures. The objects have diverse motion patterns and initial velocities. All images have dimensions of $512 \times 512$ pixels We also adopt 4 real-world scenes from HyperNeRF (Park et al., 2021) and D$^2$NeRF (Wu et al., 2022), as shown in Table 2. For the synthetic scenes, we follow D-NeRF (Pumarola et al., 2020) to employ images collected at viewpoints randomly sampled on the upper hemisphere. In contrast, real-world scenes are captured using a mobile monocular camera.

**Metrics.** For novel view synthesis, we report PSNR and SSIM (Wang et al., 2004). To compare the scene decomposition results with the 2D methods, we employ the Foreground Adjusted Rand Index (FG-ARI) (Rand, 1971; Hubert & Arabie, 1985). It measures the similarity of clustering results based on the foreground object masks to the ground truth, ranging from $-1$ to $1$ (higher is better).

Table 1: Novel view synthesis results of our approach compared with D-NeRF (Pumarola et al., 2020), DeVRF (Liu et al., 2022), and their variants to align with our experimental configurations (see text for details). We evaluate the results averaged over 60 novel views(one view per timestamp).

| METHOD | 3OBJFALL | | 3OBJRAND | | 3OBJMETAL | | 3FALL+3STILL | |
| --- | --- | --- | --- | --- | --- | --- | --- | --- |
| | PSNR↑ | SSIM↑ | PSNR↑ | SSIM↑ | PSNR↑ | SSIM↑ | PSNR↑ | SSIM↑ |
| D-NERF ($V=1$) | 28.54 | 0.946 | 12.62 | 0.853 | 27.83 | 0.945 | 24.56 | 0.908 |
| D-NERF ($V=5$) | 29.15 | 0.954 | 27.44 | 0.943 | 28.59 | 0.953 | 25.03 | 0.913 |
| DEVRF | 24.92 | 0.927 | 22.27 | 0.912 | 25.24 | 0.931 | 24.80 | 0.931 |
| DEVRF-DYN | 18.81 | 0.799 | 18.43 | 0.799 | 17.24 | 0.769 | 17.78 | 0.765 |
| OURS ($V=1$) | 31.83 | 0.967 | **31.10** | **0.966** | 29.28 | **0.954** | 27.83 | 0.942 |
| OURS ($V=5$) | **32.11** | **0.969** | 30.70 | 0.964 | **29.31** | 0.953 | **28.96** | **0.945** |
| METHOD | 6OBJFALL | | 8OBJFALL | | 3OBJREALSIMP | | 3OBJREALCMPX | |
| | PSNR↑ | SSIM↑ | PSNR↑ | SSIM↑ | PSNR↑ | SSIM↑ | PSNR↑ | SSIM↑ |
| D-NERF ($V=1$) | 28.27 | 0.940 | 27.44 | 0.923 | 27.04 | 0.927 | 20.73 | 0.864 |
| D-NERF ($V=5$) | 27.20 | 0.928 | 26.97 | 0.919 | 27.49 | 0.931 | 22.72 | 0.874 |
| DEVRF | 24.83 | 0.905 | 24.87 | 0.915 | 24.81 | 0.922 | 21.77 | 0.891 |
| DEVRF-DYN | 17.35 | 0.738 | 16.19 | 0.711 | 18.64 | 0.717 | 17.40 | 0.778 |
| OURS ($V=1$) | 29.70 | 0.948 | **29.86** | **0.945** | **30.20** | **0.952** | 26.80 | 0.917 |
| OURS ($V=5$) | **29.98** | **0.950** | 29.78 | **0.945** | 30.13 | **0.952** | **27.25** | **0.918** |

Table 2: Quantitative comparisons for novel view synthesis in real-world scenes from HyperNeRF and $D^2$NeRF. On average, our approach outperforms HyperNeRF by 5.7% in PSNR and 7.2% in SSIM. It also outperforms $D^2$NeRF, another neural renderer with disentanglement learning.

| METHOD | CHICKEN PSNR↑ | SSIM↑ | BROOM PSNR↑ | SSIM↑ | PEEL-BANANA PSNR↑ | SSIM↑ | DUCK PSNR↑ | SSIM↑ | AVG. PSNR↑ | SSIM↑ |
|---|---|---|---|---|---|---|---|---|---|---|
| NEURDIFF | 21.17 | 0.822 | 17.75 | 0.468 | 19.43 | 0.748 | 21.92 | 0.862 | 20.07 | 0.725 |
| HYPERNERF | 26.90 | **0.948** | 19.30 | 0.591 | 22.10 | 0.780 | 20.64 | 0.830 | 22.23 | 0.787 |
| $D^2$NERF | 24.27 | 0.890 | 20.66 | **0.712** | 21.35 | 0.820 | **22.07** | 0.856 | 22.09 | 0.820 |
| OURS | **27.01** | 0.934 | **21.49** | 0.702 | **24.07** | **0.863** | 21.43 | **0.878** | **23.50** | **0.844** |

Figure 3: Novel view synthesis results in synthetic and real-world scenes. D-NeRF fails in 3ObjRand. We also visualize corresponding decomposition maps for real-world scenes. It's important to note that decomposing moving objects in a real-world scene often requires additional geometric priors. Relying solely on space-time continuity within a single scene may not be sufficient to separate objects like a hand and a banana, especially when they are in contact throughout the video.

**Compared methods.** In the novel view synthesis task for synthetic scenes, we evaluate DynaVol against D-NeRF (Pumarola et al., 2020) and DeVRF (Liu et al., 2022). For real-world scenes, we compare it with NeuralDiff (Tschernezki et al., 2021), HyperNeRF (Park et al., 2021), and $D^2$NeRF (Wu et al., 2022). In the scene decomposition task, we use established 2D/3D object-centric representation learning methods, SAVi (Kipf et al., 2022) and uORF (Yu et al., 2022), as the baselines. These models are pre-trained on MOVi-A (Greff et al., 2022) and CLEVR-567 (Yu et al., 2022) respectively, which are similar to our synthetic scenes. Besides, we also include a pretrained and publicly available SAM model (Kirillov et al., 2023) to compare the segmentation performance.

## 4.2 NOVEL VIEW SYNTHESIS

**Synthetic scenes.** We evaluate the performance of DynaVol on the novel view synthesis task with the other two 3D benchmarks (D-NeRF and DeVRF). For a fair comparison, we implement "*D-NeRF (V = 5)*" which is trained using multi-view frames at the first timestamp. Besides, since DeVRF is trained with $V = 60$ views at the initial timestamp and 4 views for the subsequent timestamps, we additionally train a DeVRF model in a data configuration similar to ours. That is, we only have access to one image per timestamp ($t \geq 1$) in a dynamic view randomly sampled from the upper hemisphere. This model is termed as "*DeVRF-Dyn*". As shown in Table 1, DynaVol consistently achieves the best results in terms of PSNR and SSIM. Notably, there is no remarkable difference between the performance of our approach when trained with $V = 1$ or $V = 5$ initial images, which demonstrates that our model can be potentially applied to scenarios with monocular video data. In contrast, *DeVRF-Dyn* has a significant decline in performance compared to its standard baseline due to its heavy dependence on accurate initial scene understanding.

**Real-world scenes.** We also evaluate the performance of DynaVol in real-world scenes with the cutting-edge NeRF-based methods (NeurDiff, HyperNeRF, and $D^2$NeRF). As shown in Table 2, DynaVol outperforms the previous state-of-the-art method HyperNeRF by 5.7% in PSNR and 7.2% in SSIM on average. It also outperforms the other neural rendering method that learns disentangled representations, $D^2$NeRF, by 6.4% and 2.9% in the respective metrics. For more real-world experiments, please refer to Appendix C.

Table 3: Results in FG-ARI for object-centric scene decomposition. Since SAVi requires videos with fixed viewpoints, we generate an image sequence with a stationary camera and also evaluate DynaVol (*FixCam*) using the fixed viewpoints. Notably, unlike DynaVol, the compared models including SAM are fully generalizing models trained beyond the test scenes. In contrast, our model enables 3D scene decomposition by overfitting each individual scene in an unsupervised manner.

| METHOD | 3FALL | 3RAND | 3METAL | 3F+3S | 6FALL | 8FALL | 3SIMP | 3CMPX |
|---|---|---|---|---|---|---|---|---|
| SAVI | 3.74 | 4.38 | 3.38 | 6.12 | 6.85 | 7.87 | 3.10 | 4.82 |
| OURS (FIXCAM, $V = 5$) | **94.53** | **93.30** | **94.91** | **93.20** | **93.42** | **93.42** | **91.22** | **91.84** |
| uORF | 28.77 | 38.65 | 22.58 | 36.70 | 29.23 | 31.93 | 38.26 | 33.76 |
| SAM | 70.77 | 55.52 | 46.80 | 47.36 | 62.66 | 71.68 | 56.65 | 51.91 |
| OURS ($V = 1$) | 96.89 | **96.11** | 85.78 | 92.76 | **95.61** | 93.38 | **94.28** | 95.02 |
| OURS ($V = 5$) | **96.95** | 96.01 | **96.06** | **94.40** | 94.73 | **95.10** | 93.96 | **95.26** |

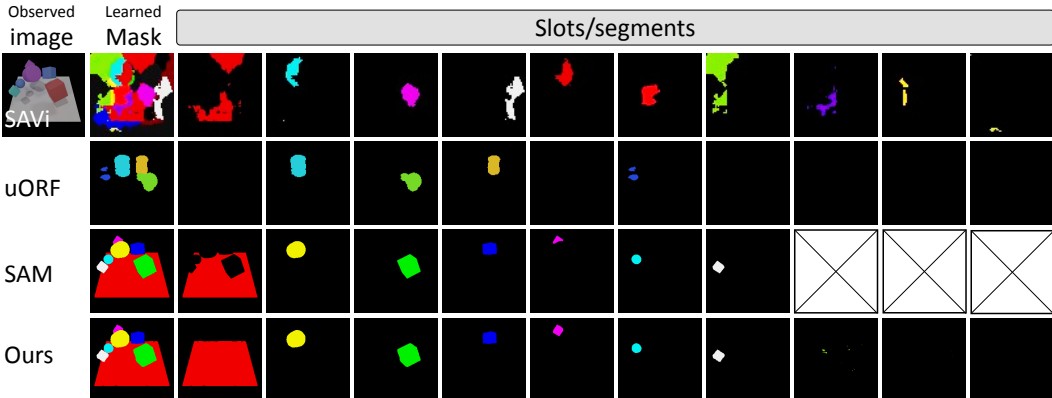

Figure 4: Visualization of scene decompostion results for *6ObjFall*.

**Qualitative results.** Figure 3 showcases the rendered images at an arbitrary timestamp from a novel view. On the synthetic dataset, it shows that DynaVol captures 3D geometries and the motion patterns of different objects more accurately than the compared methods. In contrast, D-NeRF struggles to capture intricate dynamics and fails to model the complex motion in 3ObjRand. Additionally, it generates blurry results in 3ObjRealCpmx. DeVRF, on the other hand, also fails to accurately model the trajectories of the moving object with complex textures, as highlighted by the red box in 3ObjRealCmpx. The real-world examples on the right side of Figure 3 demonstrate that DynaVol produces high-quality novel view images by understanding the stereo object-centric nature of the scene. This is further illustrated by the scene decomposition maps projected onto 2D planes.

### 4.3 SCENE DECOMPOSITION

**Quantitative results.** To obtain quantitative results for 2D segmentation as well as the scene decomposition maps, we assign the casted rays in volume rendering to different slots according to each slot's contribution to the ray's final color. Table 3 provides a comparison between DynaVol with 2D/3D unsupervised object-centric decomposition methods (SAVi/uORF) and a pretrained Segment Anything Model (SAM). For a fair comparison with SAVi, which works on 2D video inputs, we implement a DynaVol model using consecutive images with a fixed camera view (termed as *FixCam*). For uORF and SAM, as they are primarily designed for static scenes, we preprocess the dynamic sequence into $T$ individual static scenes as the inputs of these models and evaluate their average performance on the whole sequence. The FG-ARI results in Table 3 show that DynaVol ($V = 1$) and DynaVol ($V = 5$) outperform all of the compared models by large margins.

**Qualitative results.** In Figure 4, we randomly select a timestamp on 6ObjFall and present the object-centric decomposition maps of SAVi, uORF, SAM, and DynaVol. We have the following three observations from the visualized examples. First, compared with the 2D models (SAVi and SAM), DynaVol can effectively handle severe occlusions between objects in 3D space. It shows the ability to infer the complete shape of the objects, as illustrated in the 1st and 5th slots. Second, compared with the 3D decomposition uORF, DynaVolcan better segment the dynamic scene by leveraging explicitly meaningful spatiotemporal representations, while uORF only learns latent representations for each object. Last but not least, DynaVol can adaptively work with redundant slots. This means that the

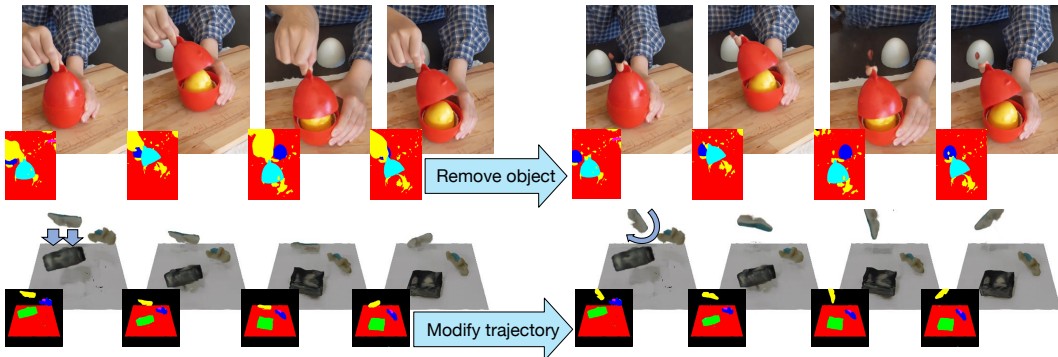

Figure 5: Showcases of dynamic scene editing for both real-world (top) and synthetic scenes (bottom). We visualize the edited decomposition map and the corresponding rendering results.

Table 4: Ablation studies of the key components and the training stages in DynaVol.

|  | 3OBJFALL | | 6OBJFALL | | 3OBJREALCMPX | |
|---|---|---|---|---|---|---|
|  | PSNR↑ | FG-ARI↑ | PSNR↑ | FG-ARI↑ | PSNR↑ | FG-ARI↑ |
| W/O FORWARD DEFORMATION $f'_\xi(t)$ | 31.50 | 95.66 | 29.75 | 92.94 | **27.35** | 94.90 |
| W/O VOLUME SLOT ATTENTION | 31.71 | 95.33 | **30.09** | 92.32 | 27.16 | 94.86 |
| W/O AVERAGE SLOTS $\bar{\mathbf{S}}^{e-1}_{1:T}$ | 31.68 | 96.19 | 29.90 | 92.97 | 27.22 | 94.90 |
| W/O 4D VOXEL OPTIMIZATION | 29.69 | 92.03 | 27.87 | 89.85 | 24.98 | 92.91 |
| 4D VOXEL OPTIM. FROM SCRATCH | 31.69 | 31.81 | 29.37 | 20.82 | 28.55 | 33.89 |
| FULL MODEL | **32.11** | **96.95** | 29.98 | **94.73** | 27.25 | **95.26** |

pre-defined number of slots can be larger than the actual number of objects in the scene. In such cases, the additional slots can learn to disentangle noise in visual observations or learn to not contribute significantly to image rendering, enhancing the model's flexibility and robustness.

## 4.4 DYNAMIC SCENE EDITING

After the training period, the object-centric voxel representations learned by DynaVol can be readily used in downstream tasks such as scene editing without the need for additional model tuning. DynaVol allows for easy manipulation of the observed scene by directly modifying the object occupancy values within the voxel grids or switching the learned deformation function to a pre-defined one. This flexibility empowers users to make various scene edits and modifications. For instance, in the first example in Figure 5, we remove the hand that is pinching the toys. In the second example, we modify the dynamics of the shoe from falling to rotating. More showcases are included in the appendix.

## 4.5 FURTHER ANALYSES

**Ablation studies.** We present ablation study results in Table 4 First, we can find that all network components are crucial to the final rendering and decomposition results. Furthermore, in the absence of the 4D voxel optimization stage, the performance of DynaVol significantly degrades, highlighting the importance of refining the object-centric voxel representation with a slot-based renderer. Additionally, we conduct an ablation study that performs 4D voxel optimization from scratch with randomly initialized $\mathcal{V}_{t=1}$ and $f_\phi(\cdot)$. The results clearly show that excluding the warmup stage has a substantial impact on the final performance, especially for the scene decomposition results.

**Analysis of the slot features.** We first study the impact of the slot number. From Table 5, we can see that find that the presence of redundant slots has only a minor impact on both rendering and decomposition results. In Figure 6, we explore the convergence of the slot values during the training

Table 5: The analysis of the impact of the number of slots ($N$) on the performance of our approach.

|  | 3OBJFALL | | 6OBJFALL | | 3OBJREALCMPX | |
|---|---|---|---|---|---|---|
| # SLOTS | PSNR↑ | FG-ARI↑ | PSNR↑ | FG-ARI↑ | PSNR↑ | FG-ARI↑ |
| 5 | 31.68 | 95.82 | 29.99 | 81.12 | 27.22 | 94.70 |
| 10 (OURS) | **32.11** | **96.95** | 29.98 | **94.73** | 27.25 | **95.26** |
| 15 | 31.71 | 95.22 | **30.51** | 84.85 | **27.27** | 94.85 |

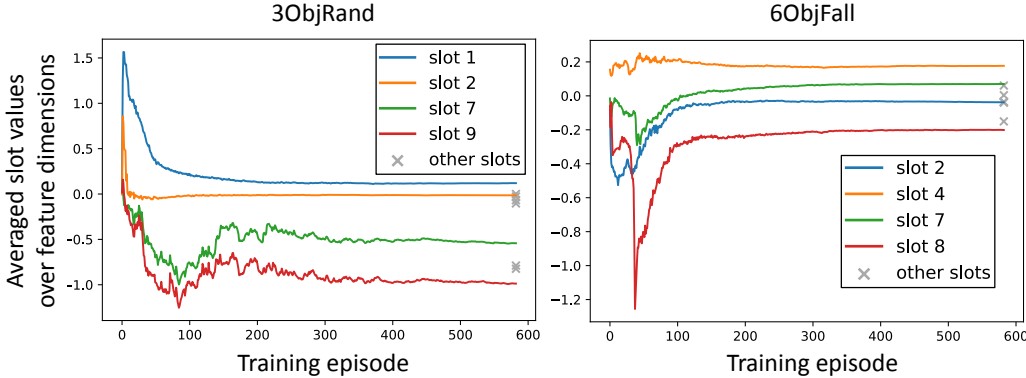

Figure 6: Visualization of the convergence of $\bar{\mathbf{S}}_{1:T}^e$ during training.

process. Specifically, we select the top 4 slots (out of 10) that contribute most to image rendering in 3ObjRand and 6ObjFall. We present the average value across all dimensions of each slot ($\{\bar{s}_n^e\}$) at different training episodes. The results demonstrate that each slot effectively converges over time to a stable value, indicating that the features become progressively refined and successfully learn time-invariant information about the scene. Furthermore, the noticeable divergence among different slots indicates that DynaVol successfully learned distinct and object-specific features.

## 5 RELATED WORK

**Unsupervised 2D scene decomposition.**    Most existing methods in this area (Greff et al., 2016; 2019; Burgess et al., 2019; Engelcke et al., 2020) use latent features to represent objects in 2D scenes. The slot attention method (Locatello et al., 2020) extracts object-centric latents with an attention block and repeatedly refines them using GRUs (Cho et al., 2014). SAVi (Kipf et al., 2022) extends slot attention to dynamic scenes by updating slots at each frame and using optical flow as the training target. STEVE (Singh et al., 2022) improves SAVi by replacing its spatial broadcast decoder with an autoregressive Transformer. SAVi++ (Elsayed et al., 2022) improves SAVi by incorporating depth information, enabling the modeling of static scenes with camera motion.

**Unsupervised 3D scene decomposition.**    Recent methods (Kabra et al., 2021; Chen et al., 2021; Stelzner et al., 2021; Yu et al., 2022; Sajjadi et al., 2022) combine object-centric representations with view-dependent scene modeling techniques like neural radiance fields (NeRFs) (Mildenhall et al., 2020). ObSuRF (Stelzner et al., 2021) adopts the spatial broadcast decoder and takes depth information as training supervision. uORF (Yu et al., 2022) extracts the background latent and foreground latents from an input static image to handle background and foreground objects separately. For dynamic scenes, Guan et al. (2022) proposed to use a set of particle-based explicit representations in the NeRF-based inverse rendering framework, which is particularly designed for fluid physics modeling. Driess et al. (2022) explored the combination of an object-centric auto-encoder and volume rendering for dynamic scenes, which is relevant to our work. However, different from our unsupervised learning approach, it requires pre-prepared 2D object segments.

**Dynamic scene rendering based on NeRFs.**    There is another line of work that models 3D dynamics using NeRF-based methods (Pumarola et al., 2020; Li et al., 2021; Liu et al., 2022; Wu et al., 2022; Guo et al., 2022; Li et al., 2023). D-NeRF (Pumarola et al., 2020) uses a deformation network to map the coordinates of the dynamic fields to the canonical space. Li et al. (2021) extended the original MLP in NeRF to incorporate the dynamics information and determine the 3D correspondence of the sampling points at nearby time steps. Li et al. (2023) achieved significant improvements on dynamic scene benchmarks by representing motion trajectories by finding 3D correspondences for sampling points in nearby views. DeVRF (Liu et al., 2022) models dynamic scenes with volume grid features (Sun et al., 2022) and voxel deformation fields. $D^2$NeRF (Wu et al., 2022) presents a motion decoupling framework. However, unlike DynaVol, it cannot segment multiple moving objects.

## 6 CONCLUSION

In this paper, we presented DynaVol, an inverse graphics method designed to understand 3D dynamic scenes using object-centric volumetric representations. Our approach demonstrates superior performance over existing techniques in unsupervised scene decomposition in both synthetic and real-world scenarios. Moreover, it goes beyond the 2D counterparts by providing additional capabilities, such as novel view synthesis and dynamic scene editing, which greatly expand its application prospects.

ACKNOWLEGMENTS

This work was supported by the National Natural Science Foundation of China (Grant No. 62250062, 62106144), the Shanghai Municipal Science and Technology Major Project (Grant No. 2021SHZDZX0102), the Fundamental Research Funds for the Central Universities, the Shanghai Sailing Program (Grant No. 21Z510202133), and the CCF-Tencent Rhino-Bird Open Research Fund.

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

APPENDIX

## A  DATA DESCRIPTION

- **3ObjFall.** The scene consists of two cubes and a cylinder. Initially, these objects are positioned randomly within the scene, and then undergo a free-fall motion along the Z-axis.
- **3ObjRand.** We use random initial velocities along the X and Y axes for each object in *3ObjFall*.
- **3ObjMetal.** We change the material of each object in *3ObjFall* from "Rubber" to "Metal".
- **3Fall+3Still.** We add another three static objects with complex geometry to *3ObjFall*.
- **6ObjFall** & **8ObjFall.** We increase the number of objects in *3ObjFall* to 6 and 8.
- **3ObjRealSimp.** We modify *3ObjFall* with real-world objects that have simple textures.
- **3ObjRealCmpx.** We modify *3ObjFall* with real-world objects that have complex textures.
- **Real-world data.** We take *Chicken*, *Peel-Banana* and *Broom* from HyperNeRF (Park et al., 2021) and *Duck* from D$^2$NeRF (Wu et al., 2022).

## B  3D-TO-4D VOXEL EXPANSION ALGORITHM

There are two steps in voxel expansion: (1) the feature graph generation and (2) the connected components computation. We input the voxel density set $\{X_k\}$ and voxel position set $\{\mathbf{x}_k\}$ with size of $N_s$, where $N_s = N_x \times N_y \times N_z$. Also, we set four parameters in advance, where $\delta_{\text{den}}$ represents the density value threshold, $\delta_{\text{rgb}}$ represents the RGB distance threshold, $\delta_{\text{vel}}$ represents the velocity distance threshold, and $N_r$ represents the number of sampling rays. By filtering out invalid locations from $\{\mathbf{x}_k\}$ with density values below a predefined threshold $\delta_{\text{den}}$, we can build a feature graph $G$ which incorporates information related to the geometry, color, and dynamics of the valid voxels. We assume that voxels belonging to the same object should have similar motion and appearance features. Conversely, voxels corresponding to objects in different spatial locations should be separated and exhibit diverse features. Subsequently, we expand the 3D voxel grid to 4D with a connected components algorithm.

## C  FURTHER EXPERIMENTS ON REAL-WORLD SCENES

To further validate DynaVol's capability on real-world scenes, we conduct additional experiments using the ENeRF-Outdoor dataset(Lin et al., 2022). As shown in Table 6, DynaVol outperforms the HyperNeRF by $10.9\%$ in PSNR and $11.0\%$ in SSIM on average. Figure 7 showcases the novel view synthesis results at an arbitrary timestamp from a novel view. Our findings reveal that DynaVol produces clearer results than HyperNeRF, especially in rendering shadows and objects held in hands. However, it's important to note that our approach falls short compared to state-of-the-art methods, such as 4K4D (Xu et al., 2023), which are based on spherical harmonics. This disparity primarily stems from different focuses in approach: the latter prioritizes renderer quality, while our emphasis lies in object-centric representation learning. Combining both approaches remains a prospect for future research.

Moreover, we showcase scene decomposition results in Figure 9, where we compare DynaVol against the state-of-the-art unsupervised video segmentation approach, OCLR (Xie et al., 2022). Additionally, OCLR employs DINO features (Caron et al., 2021) for test-time adaptation (referred to as OCLR(test adap)). Our findings indicate that DynaVol produces clearer segmentation, particularly on object boundaries, while maintaining temporal consistency. Conversely, OCLR exhibits challenges in maintaining temporal consistency. Even with test adaptation (OCLR(test adap)) it still performs not well in maintaining consistency over extended durations.

## D  IMPLEMENTATION DETAILS

We set the size of the voxel grid to $110^3$, the assumed number of maximum objects to $N = 10$, and the dimension of slot features to $D = 64$. We use 4 hidden layers with 64 channels in the renderer, and use the Adam optimizer with a batch of 1,024 rays in the two training stages. The base learning

---

**Algorithm 1** Pseudocode of the 3D-to-4D voxel expansion algorithm

---

1: **Input:** $\mathcal{X} = \{X_k\}_{k=1}^{N_s}$, $\{\mathbf{x}_k\}_{k=1}^{N_s}$, hyperparameter $\delta_{\text{den}}$, $\delta_{\text{rgb}}$, $\delta_{\text{vel}}$, $N_r$, slot number $N$
2: **Output:** $\mathcal{V}_{t=1}$
3: $\mathcal{N} = \{k | X_k > \delta_{\text{den}}, k \in [1, N_s]\}$           ▷ Filter out invalid locations
4: **for** $i \leftarrow 1$ **to** $N_r$ **do**
5:     Sample a ray randomly with direction $\mathbf{d}_i$           ▷ Get color information
6:     Calculate emitted color $\mathbf{c}_{i,j} = N_{\phi'}(\mathbf{x}_j, \mathbf{d}_i)$ **for** $j \in \mathcal{N}$
7: **end for**
8: Get averaged color $\bar{\mathbf{c}}_j = \frac{1}{N_r} \sum_{i=1}^{N_r} \mathbf{c}_{i,j}$ **for** $j \in \mathcal{N}$
9: **for** $t \leftarrow 1$ **to** $T - 1$ **do**
10:     Calculate velocity $\mathbf{v}_{t,j} = f'_\xi(\mathbf{x}_j, t+1) - f'_\xi(\mathbf{x}_j, t)$ **for** $j \in \mathcal{N}$    ▷ Get velocity information
11: **end for**
12: **for** $(p, q) \in \{(u, v) | u \in \mathcal{N}, v \in \mathcal{N}\}$ **do**
13:     $D_{\text{rgb}}(p, q) = l_2(\bar{\mathbf{c}}_p, \bar{\mathbf{c}}_q)$           ▷ Get color relationship
14:     $D_{\text{vel}}(p, q) = \max(l_2(\mathbf{v}_{t,p}, \mathbf{v}_{t,q})$ **for** $t \leftarrow 1$ **to** $T - 1)$    ▷ Get velocity relationship
15:     **if** neighbor($p$,$q$) and $D_{\text{rgb}}(p, q) < \delta_{\text{rgb}}$ and $D_{\text{vel}}(p, q) < \delta_{\text{vel}}$ **then**
16:        $E(p, q) = 1$           ▷ Create edge of the graph
17:     **else**
18:        $E(p, q) = 0$
19:     **end if**
20: **end for**
21: Generate graph feature $G$ with node $\mathcal{N}$ and edge relationship $E$
22: Obtain the cluster set $\{\text{cl}_k\}_{k=1}^M = \text{ConnectedComponents}(G)$
23: $N_l = M - N + 1$           ▷ Get number of the smallest clusters to be merged
24: $\{\text{cl}_k\}_{k=1}^N = \text{sort-and-merge}(\{\text{cl}_k\}_{k=1}^M)$    ▷ Sort clusters by size and merge $N_l$ cluster set
25: **for** $k \leftarrow 1$ **to** $N$ **do**
26:     $\mathcal{V}_{\mathbf{x}_j,k} = \mathcal{X}_j$ **for** $j \in \text{cl}_k$    ▷ Voxel expansion by assigning the density value of $\mathcal{F}$ to $\mathcal{V}$
27: **end for**

---

rates are $0.1$ for the voxel grids and $1e^{-3}$ for all model parameters in the warmup stage and then adjusted to $0.08$ and $8e^{-4}$ in the second training stage. The two training stages last for $50$k and $35$k iterations respectively. The hyperparameters in the loss functions are set to $\alpha_p = 0.1$, $\alpha_e = 0.01$, $\alpha_w = 1.0$, $\alpha_c = 1.0$. All experiments run on an NVIDIA RTX3090 GPU and last for about 3 hours.

## E  EXPERIMENTAL DETAILS OF SCENE DECOMPOSITION AND EDITING

**Scene decomposition.** To get the 2D segmentation results, we assign the rays to different slots according to the contribution of each slot to the final color of the ray. Specifically, suppose $\sigma_{in}$ is the density of slot $n$ at point $i$, we have

$$w_{in} = \frac{\sigma_{in}}{\sum_{n=0}^N \sigma_{in}}, \quad \beta_n = \sum_{i=1}^P T_i(1 - \exp(-\sigma_i \delta_i))w_{in}, \quad (5)$$

where $w_{in}$ is the corresponding density probability and $\beta_n$ is the color contribution to the final color of slot $n$. We can then predict the label of 2D segmentation by $\hat{y}(\mathbf{r}) = \text{argmax}_n(\beta_n)$.

**Scene editing.** The object-centric representations acquired through DynaVol demonstrate the capability for seamless integration into scene editing workflows, eliminating the need for additional

Table 6: Quantitative comparisons for novel view synthesis in real-world scenes from ENeRF-Outdoor. On average, our approach outperforms HyperNeRF by 10.9% in PSNR and 11.0% in SSIM.

| METHOD | ACTOR1_4 PSNR↑ | ACTOR1_4 SSIM↑ | ACTOR2_3 PSNR↑ | ACTOR2_3 SSIM↑ | ACTOR5_6 PSNR↑ | ACTOR5_6 SSIM↑ | AVG. PSNR↑ | AVG. SSIM↑ |
|---|---|---|---|---|---|---|---|---|
| HYPERNERF | 22.79 | 0.759 | 22.85 | 0.773 | 25.22 | 0.806 | 23.62 | 0.779 |
| OURS | **26.44** | **0.871** | **26.24** | **0.864** | **25.97** | **0.861** | **26.21** | **0.865** |

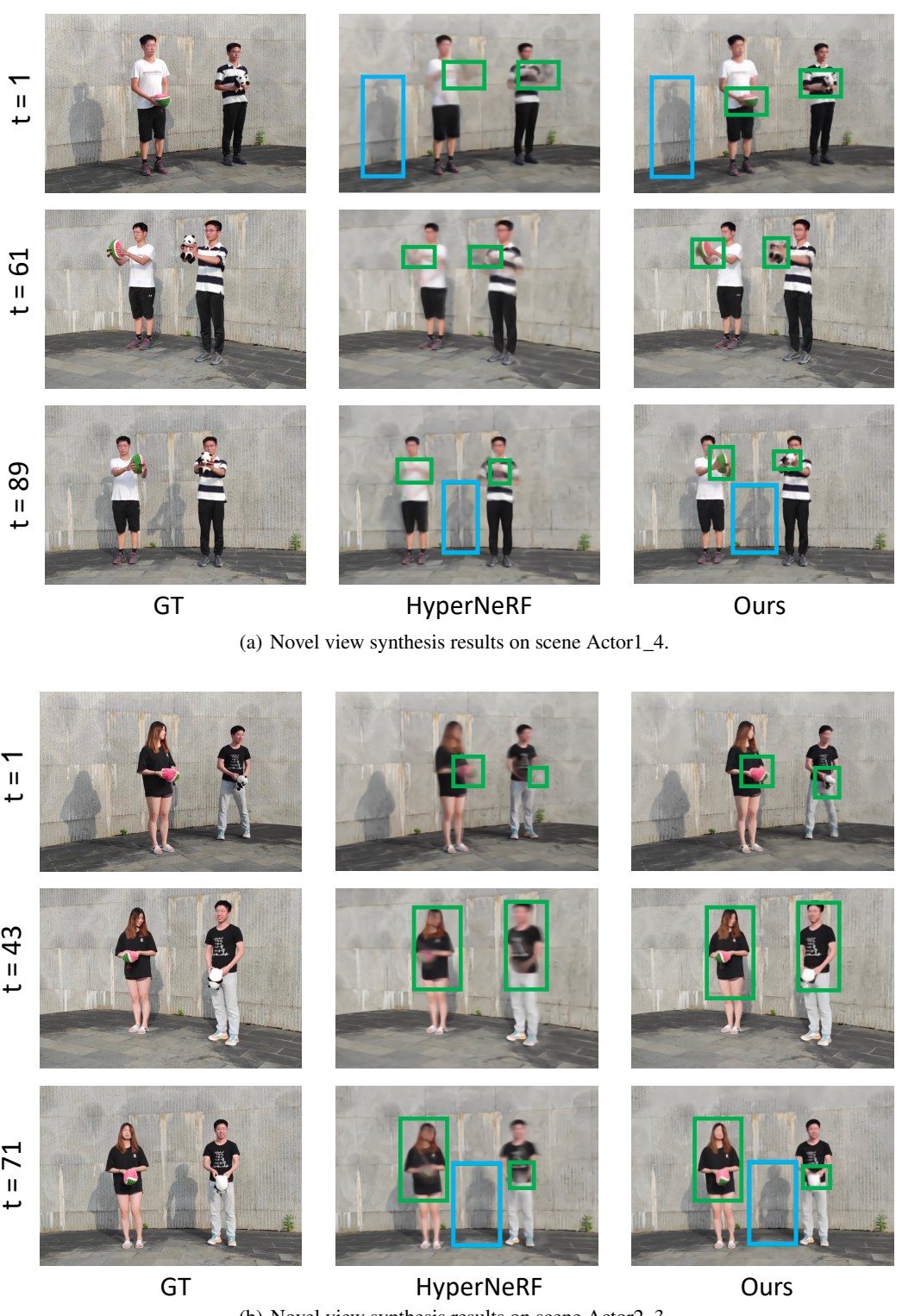

(a) Novel view synthesis results on scene Actor1_4.

(b) Novel view synthesis results on scene Actor2_3.

Figure 7: Novel view synthesis results on ENeRF-Outdoor dataset.

training. By manipulating the 4D voxel grid $\mathcal{V}_{\text{density}}$ learned in DynaVol, objects can be replaced, removed, duplicated, or added according to specific requirements. For example, we can swap the

Table 7: Hyperparameter choice of the threshold value for splitting the foreground and background in 3D-to-4D voxel expansion stage.

| | 3OBJFALL | | 6OBJFALL | | 3OBJREALCMPX | |
|---|---|---|---|---|---|---|
| THRESH | PSNR↑ | FG-ARI↑ | PSNR↑ | FG-ARI↑ | PSNR↑ | FG-ARI↑ |
| 0.1 | 31.69 | 95.73 | 29.95 | 93.08 | **27.44** | 94.37 |
| 0.001 | 31.55 | 95.64 | 29.95 | 93.15 | 27.37 | **95.33** |
| 0.01(OURS) | **32.11** | **96.95** | **29.98** | **94.73** | 27.25 | 95.26 |

Table 8: Hyperparameter choice of the weight of the $\mathcal{L}_{\text{Point}}$ loss.

| | 3OBJFALL | | 6OBJFALL | | 3OBJREALCMPX | |
|---|---|---|---|---|---|---|
| $\alpha_p$ | PSNR↑ | FG-ARI↑ | PSNR↑ | FG-ARI↑ | PSNR↑ | FG-ARI↑ |
| 0.0 | 31.49 | 95.30 | **30.46** | **95.06** | 26.84 | 95.18 |
| 1.0 | 29.90 | 89.33 | 29.25 | 93.65 | 25.49 | 55.52 |
| 0.1(OURS) | **32.11** | **96.95** | 29.98 | 94.73 | **27.25** | **95.26** |

color between objects by swapping the corresponding slots assigned to each object. Moreover, the deformation field of individual objects can be replaced with user-defined trajectories (*e.g.*, rotations and translations), enabling precise animation of the objects in the scene.

# F FURTHER EXPERIMENTAL RESULTS

**Ablation study of the hyperparameter choice.** We study the hyperparameter choice of the threshold value for splitting the foreground and background in the warmup stage in Table 7, and the weight($\alpha_p$) of per-point RGB loss in Table 8. It can be found that our model is robust to the threshold, and the performance of DynaVol remains largely unaffected with different thresholds. Furthermore, it can be found that $\alpha_p = 0.1$ performs better than $\alpha_p = 0.0$, indicating that a conservative use of $\mathcal{L}_{\text{point}}$ can improve the performance. A possible reason is that it eases the training process by moderately penalizing the discrepancy of nearby sampling points on the same ray. When the weight of $\mathcal{L}_{\text{point}}$ increases to $\alpha_p = 1.0$, the performance of the method decreases significantly. Such a substantial emphasis on $\mathcal{L}_{\text{point}}$ is intuitively unreasonable and can potentially introduce bias to the neural rendering process, thereby negatively impacting the final results.

**Ablation study of 3D volume encoder.** To derive object-level global representations from the 4D occupancy grids $V_t$, we employ the 3D volume encoder within the volume slot attention mechanism. We evaluate DynaVol without the 3D encoder in Table 9, which shows using a 3D encoder to connect the global and local object-centric features is beneficial to the performance of DynaVol.

Table 9: Ablation study of 3D volume encoder within volume slot attention.

| | 3OBJFALL | | 6OBJFALL | | 8OBJFALL | |
|---|---|---|---|---|---|---|
| METHOD | PSNR↑ | FG-ARI↑ | PSNR↑ | FG-ARI↑ | PSNR↑ | FG-ARI↑ |
| W/O 3D ENCODER | 31.26 | 96.52 | 29.90 | 93.91 | 29.21 | 93.67 |
| FULL MODEL | **32.11** | **96.95** | **29.98** | **94.73** | **29.78** | **95.10** |

**Ablation study of the slot attention steps $N_{\text{itr}}$.** The number of iterations in the slot attention is to refine the slots for more accurate object-centric features. We choose $N_{\text{itr}} = 3$ in our method, as a larger number of update steps at a single timestamp may result in the overfitting problem. Table 10 presents quantitative comparisons across various numbers of slot attention steps, revealing that DynaVol achieves optimal performance at $N_{\text{itr}} = 3$.

**Per-slot rendering results.** We present the per-slot rendering results in Figure 8, demonstrating DynaVol's robustness to varying slot numbers. Our findings indicate that redundant slots remains empty, thus mitigating the issue of over-segmentation.

Table 10: Ablation study of the slot attention steps $N_{\text{itr}}$ repeated at each timestamp.

| $N_{\text{ITR}}$ | 3OBJFALL | | 6OBJFALL | | 8OBJFALL | |
|---|---|---|---|---|---|---|
| | PSNR↑ | FG-ARI↑ | PSNR↑ | FG-ARI↑ | PSNR↑ | FG-ARI↑ |
| 1 | 31.39 | 96.26 | 29.90 | **94.88** | 29.58 | 93.65 |
| 5 | 31.59 | 96.42 | 29.65 | 94.77 | 29.69 | 93.55 |
| 3(OURS) | **32.11** | **96.95** | **29.98** | 94.73 | **29.78** | **95.10** |

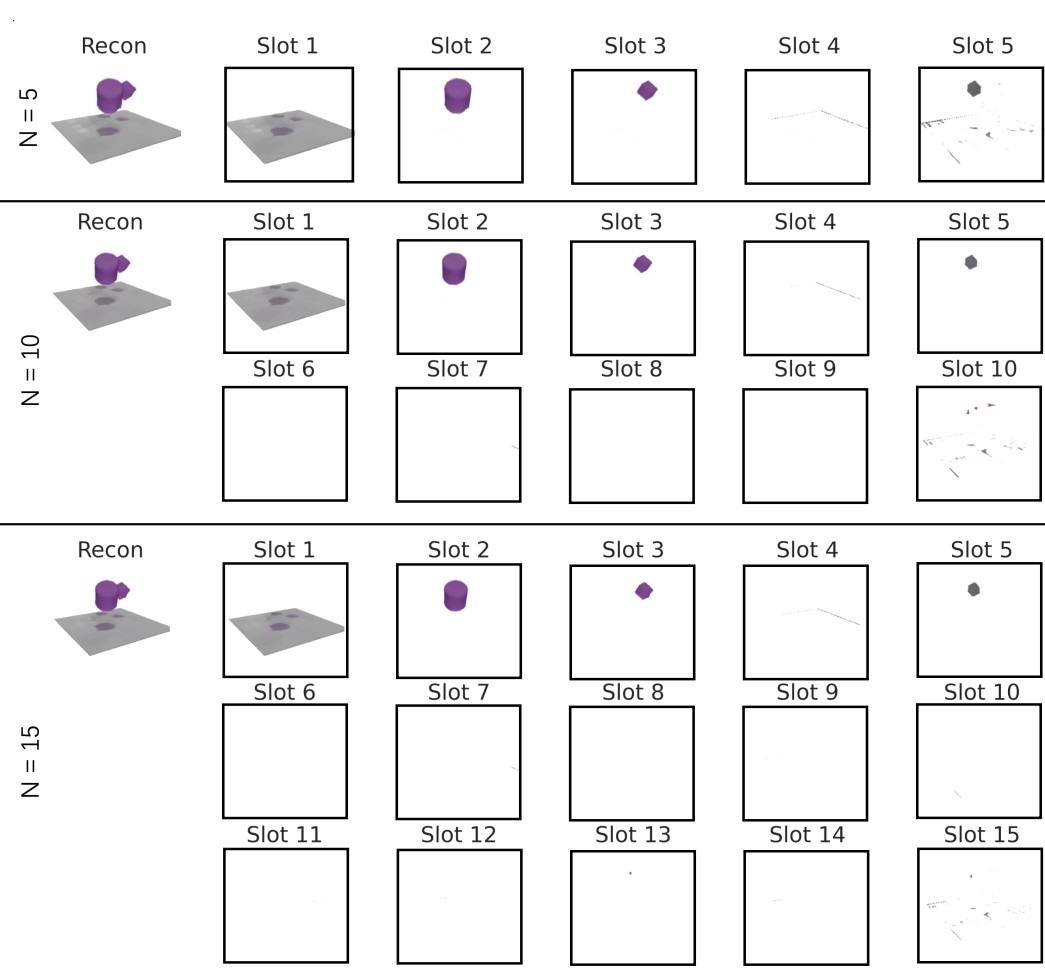

Figure 8: Per-slot rendering results for scene *3ObjFall* with 5, 10, and 15 slots, respectively.

**Error bars.** To assess the performance stability of DynaVol, we perform separate training processes using three distinct seeds. The results, presented in Table 11, showcase the mean and standard deviation of the PSNR (Peak Signal-to-Noise Ratio) and FG-ARI (Adjusted Rand Index for Foreground) values for novel view synthesis and scene decomposition tasks. These additional results serve as a valuable supplement to those presented in Table 1 and Table 3 in the main manuscript, demonstrating the consistent and reliable performance of DynaVol across multiple training trials.

## G    ENLARGED VISUALIZATION OF EDITED DYNAMIC SCENES

In Figures 10–13, we provide enlarged visualizations of the edited dynamic scenes for better clarity and observation. These figures provide a closer look at the specific changes made to the dynamic scenes, enabling a better understanding of the editing process.

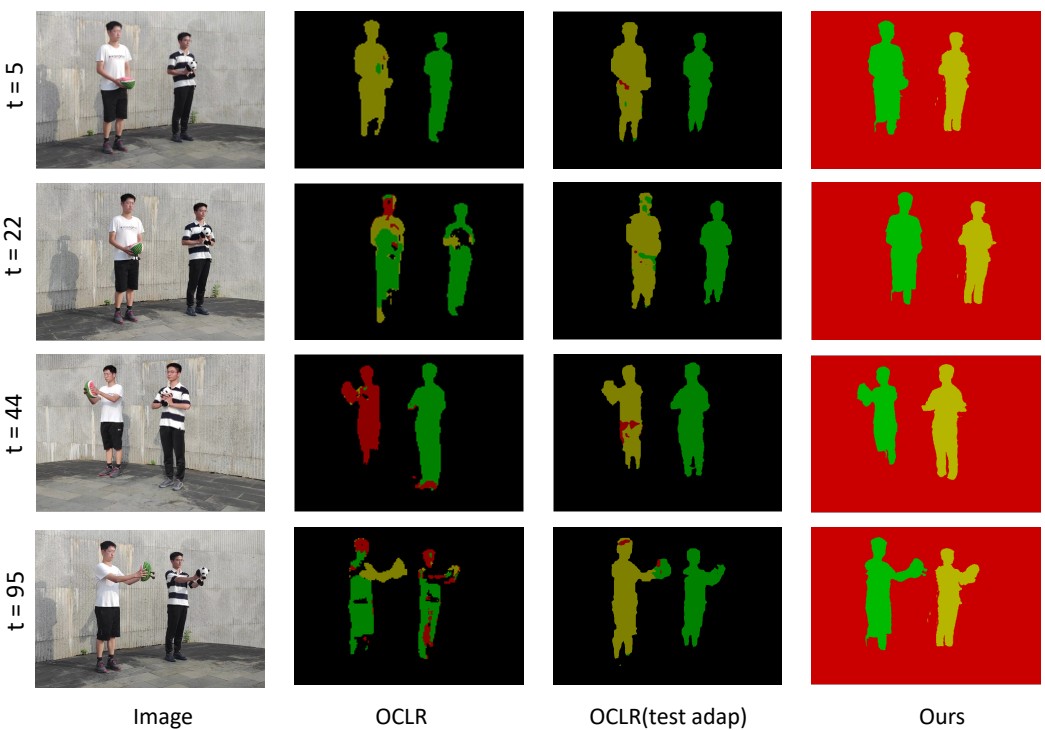

Figure 9: Scene decomposition results on ENeRF-Outdoor dataset.

Table 11:  Quantitative results with error bars (mean± std) of DynaVol.

| 3ObjFall | | 3ObjRand | | 3ObjMetal | | 3Fall+3Still | |
|---|---|---|---|---|---|---|---|
| PSNR | FG-ARI | PSNR | FG-ARI | PSNR | FG-ARI | PSNR | FG-ARI |
| 32.05±0.06 | 97.03±0.08 | 30.79±0.11 | 96.04±0.04 | 29.37±0.07 | 96.02±0.11 | 28.97±0.02 | 94.36±0.06 |
| 6ObjFall | | 8ObjFall | | 3ObjRealSimp | | 3ObjRealCmpx | |
| PSNR | FG-ARI | PSNR | FG-ARI | PSNR | FG-ARI | PSNR | FG-ARI |
| 29.99±0.07 | 94.75±0.17 | 29.79±0.06 | 95.12±0.03 | 30.17±0.07 | 94.10±0.13 | 27.14±0.10 | 95.23±0.05 |

For more examples, including object adding and duplication, please refer our project page: `https://sites.google.com/view/dynavol/`, which demonstration showcases further instances of scene editing using DynaVol, providing a comprehensive overview of its capabilities.

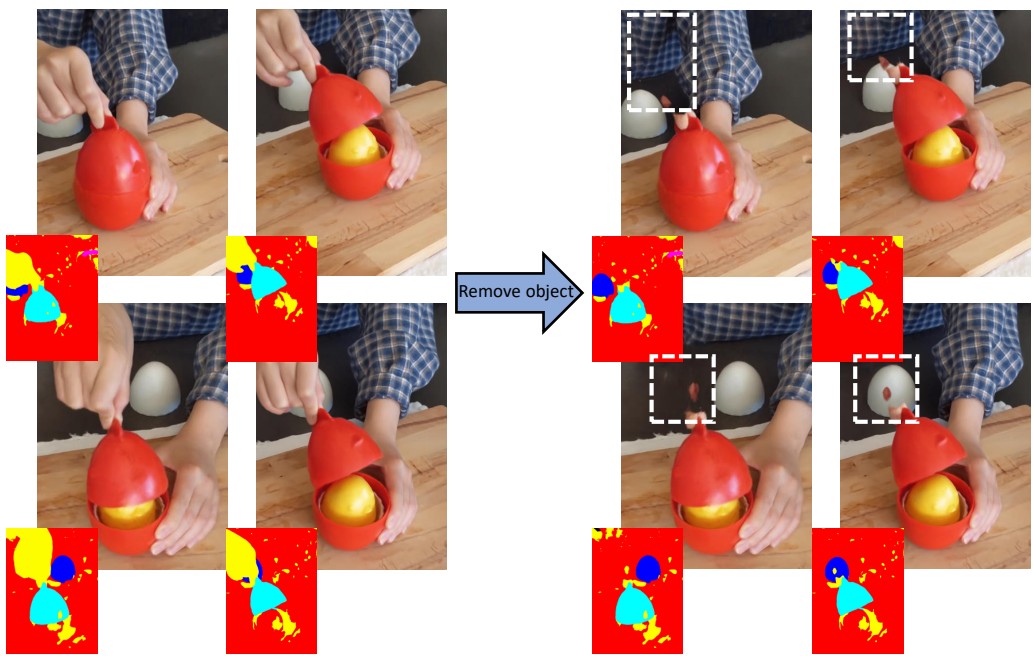

Figure 10: Real-world decomposition and scene editing. The left images illustrate the original scene before any editing, while the right images present the scene after object removal. It is an enlarged version of Figure 5 in the main manuscript.

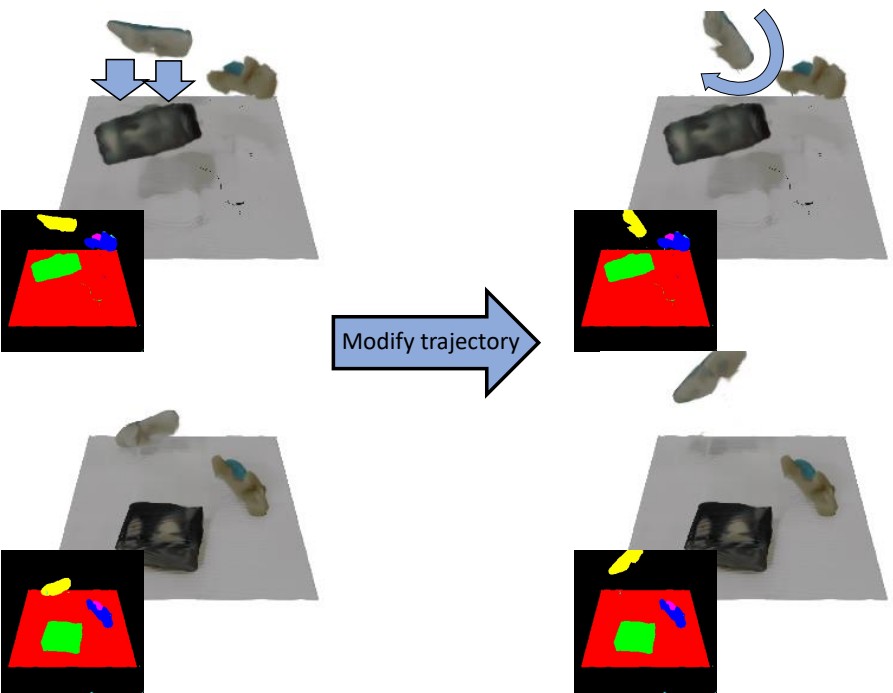

Figure 11: Trajectory modification based on *3ObjRealCmpx* (Left: before editing; Right: after editing). It is an enlarged version of Figure 5 in the main manuscript.

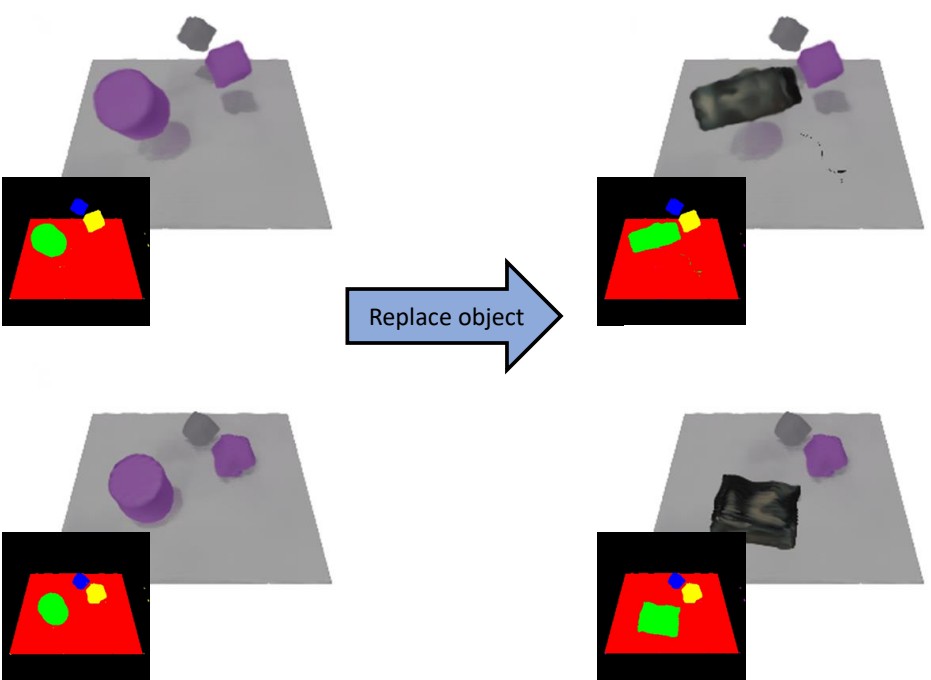

Figure 12: Object replacement based on *3ObjFall* and *3ObjRealCmpx* (Left: before editing; Right: after editing).

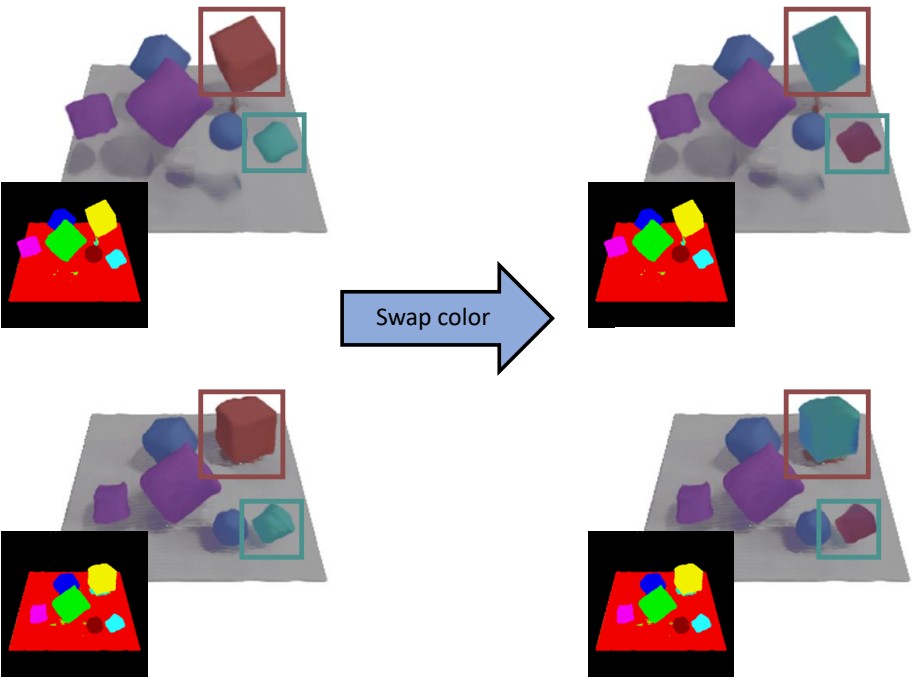

Figure 13: Color swapping based on *6ObjFall* (Left: before editing; Right: after editing).

