# OpenReview forum: "DynaVol: Unsupervised Learning for Dynamic Scenes through Object-Centric Voxelization"
_ICLR.cc/2024/Conference — ICLR 2024 poster_

### Official Review · Reviewer_p1WX · 2023-10-28

**Soundness:** 3 good
**Presentation:** 2 fair
**Contribution:** 2 fair
**Rating:** 3
**Confidence:** 3

**Summary:**

This paper presents an unsupervised learning method for 3D dynamic scene capture and decomposition. The presented method, DynaVol, utilizes a canonical NeRF as well as a time-conditioned implicit warping field for modeling a 3D dynamic scene. Learning such a representation uses a monocular video together with multi-view observation of a static frame, and several other objectives designed for retrieving better geometry and enforcing temporal consistency. Comparisons with previous state-of-the-arts are shown on the tasks of novel view synthesis and 3D scene decomposition, and improved results are reported.

**Strengths:**

* Experiment

 Extensive experimental results are shown on tasks like novel view synthesis and 3D scene decomposition. Improved results are achieved compared to many existing baselines and state-of-the-arts.

**Weaknesses:**

* More insights from ablation study?

It is really good to see that the authors conduct a lot of ablation study on the work. However, some of them only consist of a chart with statistics, while lacking some insight analysis on them. It would further improve the quality of this work if more insights are discussed there.

* Novelty?

The method proposed in the manuscript is pretty interesting, although it would be great to highlight the novelty and difference between the proposed method and previous literature. For example, on the dynamic modeling side, what is the relationship with [1, 2]. On the representation side, the design of volume slot attention is also not very well justified. It would be great if more insights and ablations are provided for the that part of design.

* Introduction

In the introduction part, I feel the importance of the problem that the paper is working on is not well explained. Instead, the author directly starts to focus on what method is proposed and what experiments are conducted. It would better enhance the quality of the paper if more background and impact of solving this problem are provided and discussed.

[1] Li, Zhengqi, et al. "Neural scene flow fields for space-time view synthesis of dynamic scenes." Proceedings of the IEEE/CVF Conference on Computer Vision and Pattern Recognition. 2021.
[2] Li, Zhengqi, et al. "Dynibar: Neural dynamic image-based rendering." Proceedings of the IEEE/CVF Conference on Computer Vision and Pattern Recognition. 2023.

**Questions:**

* Supervision?

Although not a very big concern, the number of objects N seems to serve as input to the pipeline. This semantic could serve as additional supervision with strong prior, especially to the task of scene decomposition. It would be interesting to see how to address the problem of automatically determining the number of objects in the scene.

---

> ### Author Response · Authors · 2023-11-22
>
> Thank you for your comments. Please find our responses to each of them below. If you have any further concerns or questions, please feel free to inform us.
>
> > Q1. More insights from ablation study.
>
> We appreciate your suggestion and have incorporated more discussion regarding the ablation study results in Section 4.5 of the revised paper. Specifically, we delve into the impact of different stages during the training process:
> - **Warmup Stages:** We conducted an additional ablation study performing 4D voxel optimization from scratch with randomly initialized $\mathcal{V}{t=1}$ and $f_\phi(\cdot)$. This study clearly illustrates that excluding the warmup stages (which includes both the "3D voxel warmup stage" and the "3D-to-4D voxel expansion stage") has a substantial impact on the final performance, particularly for the scene decomposition results.
> - **4D Voxel Optimization Stage:** We emphasize that the performance of DynaVol significantly degrades when excluding the 4D voxel optimization stage. These results demonstrate the importance of refining the object-centric voxel representation with a slot-based renderer.
>
> > Q2. The method proposed in the manuscript is pretty interesting, although it would be great to highlight the novelty and difference between the proposed method and previous literature.
>
> We have included more discussion of the differences and relationships between our model and the suggested existing work [1,2] in Section 5 in the revision. We here provide a more detailed comparison of these methods on the dynamic modeling side:
> - **DynaVol:** In addition to the rendering MLP, our model employs additional networks that learn $(\mathbf{x},t) \rightarrow \Delta \mathbf{x}\in \mathbb{R}^{3}$ to explicitly transport the density values in each voxel grid to the corresponding voxel grid at an arbitrary future time step.
> - **Method from [1]:** This approach extends the original rendering MLP in NeRF to incorporate the dynamics information. Basically, it learns a mapping function of (RayPoint, ViewDirection, Time) $\rightarrow$ (Color, Density, Offset). Here, "offset" determines the 3D correspondence of the RayPoint at nearby time steps.
> - **Method from [2]:** This approach has achieved significant improvements on dynamic scene benchmarks by representing motion trajectories by finding 3D correspondences for sampling points in nearby views.
>
> Furthermore, while all of these methods leverage temporal-consistency constraints, unlike [1,2], which establish the constraint in pixel color space, we incorporate it in the canonical space to encourage coherent transportations of the density values across time.

---

> > ### Author Response · Authors · 2023-11-22
> >
> > > Q3. On the representation side, the design of volume slot attention is also not very well justified. It would be great if more insights and ablations were provided for that part of the design.
> >
> > The key insight of volume slot attention is to progressively extract a set of global object-level representations from the local voxel grid features. In the original manuscript (Table 4), we have showcased the significance of the entire slot attention module by replacing it with a set of learnable variables. In the revised paper, we have introduced a new ablation study (Table 4) and further clarified the main ideas behind specific model designs of the volume slot attention module (Section 3.3). Here are some details:
> >
> > (1) For the 3D volume encoder:
> > - Insight: To extract object-level global representation from the 4D occupancy grids $V_t$.
> > - Ablation study (**New!**): Below, we evaluate our model without the 3D encoder, which shows using a 3D encoder to connect the global and local object-centric features is beneficial to the performance of DynaVol in PSNR and ARI-FG.
> >
> > |        |      3ObjsFall      |      6ObjsFall      |      8ObjsFall      |
> > | ------------------- |:-------------------:|:-------------------:|:-------------------:|
> > | w/o 3D Encoder |       31.26 $\mid$ 96.52   |    29.90 $\mid$ 93.91      |   29.21 $\mid$ 93.67     |
> > | Full model          | **32.11** $\mid$ **96.95** | **29.98** $\mid$ **94.73** | **29.78** $\mid$ **95.10** |
> >
> > (2) For using the averaged slot features over the sequence to drive the renderer:
> > - Insight: To allow the rendering MLP to be conditioned on time-invariant representations of each object, such as color and materials.
> > - Ablation study: It has been included in the original Table 4.
> >
> > (3) For the slot attention steps $N\_\text{itr}$ repeated at each timestamp $t$:
> > - Insight: The number of iterations in the slot attention is to refine the slots for more accurate object-centric features. We choose $N_\text{itr}=3$ in our method, as a larger number of update steps at a single timestamp may result in the overfitting problem.
> > - Ablation study (**New!**): Below, we provide the quantitative comparisons using different numbers of the slot attention steps. DynaVol achieves the best performance in PSNR and ARI-FG at $N\_\text{itr}=3$.
> >
> > |                  |      3ObjsFall      |    6ObjsFall    |      8ObjsFall      |
> > | ---------------------- |:-------------------:|:---------------:|:-------------------:|
> > | $N_\text{itr}=1$       |     31.39 $\mid$ 96.26     | 29.90 $\mid$ **94.88** |     29.58/93.65     |
> > | $N_\text{itr}=3$ (Ours) | **32.11** $\mid$ **96.95** | **29.98** $\mid$ 94.73 | **29.78**/**95.10** |
> > | $N_\text{itr}=5$       | 31.59 $\mid$ 96.42 | 29.65 $\mid$ 94.77 | 29.69 $\mid$ 93.55 |
> >
> > >  Q4. Introduction: It would better enhance the quality of the paper if more background and the impact of solving this problem were provided and discussed.
> >
> > We greatly appreciate the reviewer's suggestion and have revised the introduction accordingly. The importance of our work lies in three aspects:
> > * **A New Problem definition:** DynaVol aims to solve unsupervised object decomposition in a 3D dynamic scene with images collected by a monocular camera. It is a new research problem in computer vision.
> > * **Comparision to Previous Art:** Most existing approaches for unsupervised dynamic scene decomposition (such as SAVi++) as well as the models with large-scale supervised pretraining (such as SAM) ignore the 3D nature of the scene. In comparison, we present the first model for unsupervised 3D dynamic scene decomposition that naturally ensures 3D-consistent scene decomposition.
> > * **Impact on Downstream Tasks:** In addition to the ability to synthesize novel views and decompose 3D dynamic scenes, the learned object-centric occupancy grids in DynaVol can be easily used in downstream tasks such as scene editing (which is not supported by the 2D counterparts).

---

> > > ### Author Response · Authors · 2023-11-22
> > >
> > > > Q5. (1) The number of objects N seems to serve as input to the pipeline. This semantic could serve as additional supervision with strong prior, especially to the task of scene decomposition. (2) It would be interesting to see how to address the problem of automatically determining the number of objects in the scene.
> > >
> > > (1) Clarification of the meaning of $N$:
> > >
> > > First, we would like to clarify that $N$ represents **the number of slots** rather than the number of objects (sorry for the misleading words in the paper). In all experiments, $N$ is manually pre-defined as 10 and is assumed to be larger than the number of objects in the scene. In other words, **we do not know the exact number of objects**. We have clarified the meaning of $N$ in the revised paper.
> > >
> > > Notably, using redundant slots naturally mitigates the *under-segmentation* problem. As shown in Figure 4 in the main text (*Per-slot decomposition results*) and Figure 8 in Appendix.F (**New!** *Per-slot rendering results*), the redundant slots can adaptively focus on capturing the background noises in the visual scene without making a substantial contribution to the final rendering results.
> > >
> > > Additionally, this is a common practice to assume a known maximum value for the number of objects in the scene and set the number of slots to be larger than the number of objects. This setting is consistent with prior work, including Slot Attention, SAVi, SAVi++, etc.
> > >
> > > (2) Automatically determining $N$ in the scene:
> > >
> > > As suggested by the reviewer, we have tried to adaptively determine the value of $N$ using the connected components algorithm. Here are a few steps in this procedure:
> > > - *Initial Filtering:* Following the initial warmup stage, we filter out empty voxels with low-density values.
> > > - *Connected Component Identification:* We apply Algorithm 1 in the appendix of the revised paper to obtain $M$ connected components. However, $M$ tends to be larger than the number of objects in the scene.
> > > - *Voxel Contribution Calculation:* To filter out noisy connected components, we compute the contributions of each voxel (denoted by $m_i$) to the rendering results: $m_i =T_i \cdot \alpha_i$, where $\alpha_i=1-\text{exp}(-\sigma_i \delta_i)$ represents the ray termination probability with volume density $\sigma_i$ and the distance $\delta_i$ between adjacent sampling points along the ray. Additionally, $T_i=\prod_{j=1}^{i-1}(1-\alpha_j)$ represents the accumulated transmittance from the camera position to the sampling point $i$. This approach aligns with the work of "*Compressing volumetric radiance fields to 1 MB*" by Li et al. (2023).
> > > - *Accumulation and Sorting:* We accumulate the $m_i$ over all voxels in each connected component and sort the accumulated contribution values of the connected components.
> > > - *Obtaining Top-$N$ Components:* We filter out the connected components with lower accumulated contribution values to ensure that the sum of $m_i$ of the remaining voxels is larger than 95% of that of all non-empty voxels. This process ultimately determines the number of slots, $N$, as the number of remaining connected components.
> > >
> > > **Quantitative Comparison:** We compare the results obtained using the adaptively determined value of $N$ with those using the manually pre-defined value of $N$. The results are presented in terms of PSNR and FG-ARI, respectively:
> > >
> > > | How to determine the number of slots? |       3ObjsFall (#Objects=4)        |     6ObjsFall (#Objects=7)      |       8ObjsFall (#Objects=9)        |
> > > | --------------------- |:-----------------------------------:|:-------------------------------:|:-----------------------------------:|
> > > | Manually pre-defined  | **32.11** $\mid$ **96.95** ($N=10$) | **29.98** $\mid$ 94.73 ($N=10$) | **29.78** $\mid$ **95.10** ($N=10$) |
> > > | Adaptively determined |     31.68 $\mid$ 95.82 ($N=5$)      | 29.85 $\mid$ **95.29** ($N=8$)  |     29.51 $\mid$ 93.16 ($N=9$)      |
> > >
> > > **Conclusion:** Our results suggest that manually pre-defining the number of slots can achieve comparable performance to the adaptive approach mentioned above, with the added benefit of being significantly easier to implement.

---

### Official Review · Reviewer_FQjp · 2023-10-31

**Soundness:** 3 good
**Presentation:** 2 fair
**Contribution:** 3 good
**Rating:** 6
**Confidence:** 3

**Summary:**

The manuscript proposes a method, DynaVol, that learns to represent dynamic scenes as decomposed into the different moving parts (objects). The input to the method is a sequence of frames and their pose.
This decomposition is learned in 3D by learning to deform a canonical volume into the 3d states at the different observation points. Slot attention on 3d voxel grids is used to learn time-invariant object representations that are deformed by the deformation field. A NERF rendering loss guides allows learning this from just the input image sequence.

**Strengths:**

The research direction of learning to decompose and represent a dynamic scene in terms of the motion of individual constituent parts (objects / slots) is important.

The results on simulated data are encouraging (although there are only a few scenes shown) and show cases where the model is indeed able to nicely separate out the different moving parts of the scene.
Figure 4 exactly captures the advantage of the proposed dynamic object-centric scene representation in comparison to 2d methods (SAM): being able to represent unobservable parts of an image.

The proposed approach to dynamic scene representation has fundamental advantages as demonstrated by the qualitative results on scene editing (removing objects; modifying dynamics).

The visuals and writing in the manuscript are of high quality (the clarity could be improved - see weaknesses).

**Weaknesses:**

The proposed model DynaVol is complex. The text and the architecture figures were not easy to follow. Especially the parts about the slot updating and training are a bit opaque to me still. There are various unexplained variables in Fig. 1 which means the figure doesnt really add much to explaining the concepts and architecture to me.

One weakness is that DynaVol has to be overfited to each individual single scene. As such this is similar to NERF and okay. But when comparing such an overfit model to a fully generalizing model like SAM caution has to be use to clarify very precisely.

The experiments do suggest to me that DynaVol works well on easily segmented objects (from the simulation) but on real sequences DynaVol does not much outperform other methods. The small set of sequences evaluated in sim and real weakens the claims.

**Questions:**

Do I understand correctly that there is only one occupancy grid at t=0 and the other ts are reconstructed using the deformation field? A query at time t is warped back to t=0 and then interpolated into the feature volume?

Where is the GRU in Fig 2?

For the NERF rendering is it correct that the occupancy comes from the occupancy grid and the color comes from the slot features? Why this choice and not push occupancy into the slot features too?

I couldnt find the per-point color loss in the DVGO paper? What am I missing?

What happens without the warmup phase?

How are the occupancy grid and the deformation function represented? Just dense values on a grid? what resolution?

How is the feature graph computed after warmup?

---

> ### Author Response · Authors · 2023-11-22
>
> We appreciate your great efforts in reviewing our paper and hope that the following responses can address most of your concerns.
>
> > Q1. The proposed model DynaVol is complex. The text and the architectural figures were not easy to follow.
>
> We apologize for any difficulty you experienced in reading our paper. We have tried our best to organize the writing but found it challenging due to the complexity of the proposed method, which involves multiple training stages and many details. To improve the overall reading experience, we have made the following changes:
> - We refined Section 3.1 and Figure 2 for a better overview of DynaVol.
> - We revised Section 3.4 to reorganize the entire training pipeline and separate it into three stages, including a "3D voxel warmup stage", a "3D-to-4D voxel expansion stage", and a "4D voxel optimization stage". We further clarified the key insight of each stage and the connections between these stages.
> - We introduced Algorithm 1 in Appendix.B to present the details of 3D-to-4D voxel expansion, outlining how to initialize the 4D voxel representation using the connected component algorithm.
>
> Please refer to the updated paper for the details.
>
> > Q2. When comparing such an overfit model to a fully generalizing model like SAM caution has to be used to clarify very precisely.
>
> Thanks for the suggestion! We have clarified the differences in the experimental setups of DynaVol and SAM in the caption of Table 3: "*The compared models, including SAM, are fully generalizing models trained beyond the test scenes. In contrast, our model enables 3D scene decomposition by overfitting each individual single scene in an unsupervised manner.*" Despite the distinct training setups, the comparison in Table 3 aims to showcase DynaVol's ability for 3D scene decomposition by leveraging knowledge of the segmented scene.
>
> Note: Both the quantitative results in Table 3 and the qualitative results in Figure 4 correspond to images of novel views outside the scope of DynaVol's training set.
>
> > Q3. DynaVol works well on easily segmented objects (from the simulation) but on real sequences, DynaVol does not much outperform other methods.
>
> We have added new real-world experiments on the ENeRF-Outdoor dataset. Please refer to Figure 7 in Appendix.C for the qualitative results. Below, we present the quantitative comparisons in PSNR and SSIM:
>
> |  Real-world scenes  |       Actor1_4        |     Actor2_3      |
> | --------------------- |:-----------------------------------:|:-------------------------------:|
> | HyperNeRF  | 22.79 $\mid$ 0.759 | 22.85 $\mid$ 0.773 | 25.22 $\mid$ 0.806|
> | Ours |    **26.44** $\mid$ **0.871**      |  **26.24** $\mid$ **0.864** |
>
> > Q4. Do I understand correctly that there is only one occupancy grid at t=0 and the other ts are reconstructed using the deformation field? A query at time t is warped back to t=0 and then interpolated into the feature volume?
>
> Yes.
>
> > Q5. Where is the GRU in Fig 2?
>
> We consider GRU as a component of slot attention, thereby placing it in $Z\_\omega$. We clarified this in the caption and the legend in Figure 2.
>
> > Q6. For the NERF rendering is it correct that the occupancy comes from the occupancy grid and the color comes from the slot features? Why this choice and not push occupancy into the slot features too?
>
> Yes, as illustrated in the updated Figure 2, we determine the color $\bar{\mathbf{c}}$ at a ray sampling point through the following steps, all starting from the occupancy grids $\mathcal{V}\_t$:
> - $\mathcal{V}\_t$ $\rightarrow$ [Volume Slot Attention] $\rightarrow$ Slot features $s_t^n$ $\rightarrow$ [MLP] $\rightarrow$ Per-slot color $c\_n$,
> - $\mathcal{V}\_t$ $\rightarrow$ [Fetch and Trilinear Interpolation] $\rightarrow$ Per-slot occupancy $\sigma\_n$,
> - $(c\_n, \sigma\_n)$ $\rightarrow$ [Accumulated as Eq. (3)] $\rightarrow \bar{\mathbf{c}}$.
>
> Representing per-object volume density through explicit occupancy grids offers three advantages:
> - It speeds up the training process.
> - It improves the convenience of the downstream scene editing applications.
> - It allows seamless integration with off-the-shelf scene decomposition algorithms such as the connected components.

---

> ### Author Response · Authors · 2023-11-22
>
> > Q7. Could not find the per-point color loss in the DVGO paper.
>
> Yes, indeed, DVGO did not introduce the formulation of the per-point RGB loss in the main text, but examined its effectiveness in Table I6 in the supplementary material (please refer to its arxiv version).To improve clarity, we have presented the formulation of this loss term in Eq. (5) in our original manuscript.
>
> > Q8. What happens without the warmup phase?
>
> To answer this question, we conducted an ablation study by training the entire model from scratch --- We directly performed "4D voxel optimization" with randomly initialized $\mathcal{V}\_{t=1}$ and $f\_\phi(\cdot)$. The obtained results in terms of PSNR and FG-ARI (higher values are favorable) are presented below. It is evident that removing the warmup stage significantly impacts the final performance, especially for the scene decomposition results. We also include these comparisons in Table 4 ("4D voxel optim. from scratch") in the revised paper.
>
> |   |         3ObjsFall          |         6ObjsFall          |         8ObjsFall          |
> | ---------- |:--------------------------:|:--------------------------:|:--------------------------:|
> | w/o Warmup |     31.69 $\mid$ 31.81     |     29.37 $\mid$ 20.82     |     28.55 $\mid$ 33.89     |
> | Full model | **32.11** $\mid$ **96.95** | **29.98** $\mid$ **94.73** | **29.78** $\mid$ **95.10** |
>
> A side note: We have reorganized the entire training scheme for better clarity. Hence, the "w/o Warmup" results in the above table are obtained by excluding the first two stages below:
> - *3D voxel warmup stage*: Corresponds to the training phase in the original warmup stage.
> - *3D-to-4D voxel expansion stage*: Corresponds to the post-processing part of the original warmup stage (i.e., the computation of connected components).
> - *4D voxel optimization stage*: Corresponds to the dynamic scene optimization stage in the original manuscript.
>
> > Q9. How are the occupancy grid and the deformation function represented? Just dense values on a grid? what resolution?
>
> The resolution of the occupancy grids is $\mathcal{V}\_{density}^{t=1} \in \mathbb{R}^{N \times N\_x \times N\_y \times N\_z}$, where $N$ is the number of slots, and $N\_x$, $N\_y$, and $N\_z$ represent the spatial resolution along each axis. Specifically, we set the spatial resolution of the voxel grids to $110^3$.
>
> As for the deformation function, we employ an MLP that takes $(\mathbf{x},t)$ with positional encoding as inputs to predict the position movement $\Delta \mathbf{x} \in \mathbb{R}^{3}$.
>
> > Q10. How is the feature graph computed after warmup?
>
> We have included **Algorithm 1** in Appendix B to provide a detailed explanation of the 3D-to-4D voxel expansion process, which consists of the following two steps:
> - **Feature Graph Generation:** We consider the voxel set $\\{X_k\\}\_{k=1}^{N_x \times N_y \times N_z}$ from $\mathcal{F}\_{t=1}$. By filtering out invalid locations with density values below a pre-defined threshold (inspired by DVGO), we can get the node set $\\{X_k\\}\_{k=1}^{K}$ to build the feature graph $G$. For any pair of nodes $(u,v)$ in $G$, an edge is defined between them if the following three conditions are satisfied:
>     - $u$ and $v$ correspond to spatially adjacent voxels.
>     - $u$ and $v$ result in similar emitted color $(c\_u^r, c\_v^r)$ produced by $N_{\phi^\prime}$ along an arbitrary ray $r$, such that $\frac{1}{\mathcal{R}}\sum\_r^\mathcal{R} d_\text{rgb}(c\_u^r, c\_v^r)<\text{Threshold}\_\text{rgb}$, where $\mathcal{R}$ is the number of rays randomly sampled with different directions. We take the Euclidean distance to calculate $d\_\text{rgb}(c\_u^r, c\_v^r)$.
>      - $u$ and $v$ lead to similar velocities across all time steps, which can be acquired through the forward deformation network $f^\prime\_\xi$, such that $\text{max}\_t(d_\text{vel}(u,v)) < \text{Threshold}\_\text{vel}$, where we take the maximum Euclidean distance $||(f^\prime\_\xi(u,t)-f^\prime\_\xi(u,t-1))-(f^\prime\_\xi(v,t)-f^\prime\_\xi(v,t-1))||_2$ over all time steps ($t \in [2,T]$).
>
> * **Connected Components Computation:** In this step, we first feed the feature graph $G$ into the connected components algorithm to obtain $M$ clusters. We then sort the clusters by size and merge the smallest $M-N+1$ clusters. For each cluster $p$ ($1 \le p \le N$), we assign the density value of voxels in $\mathcal{F}\_{t=1}$ to the corresponding channel at the same localation in $\mathcal{V}\_{t=1}$.

---

> ### Comment · Reviewer_FQjp · 2023-11-22
>
> Thank you for the clarifications. A lot of things are more clear now. I appreciate the additional ablations of no warmup. The real world experiment in the appendix is also nice and clean.
>
> Re Q1: 3D voxel warmup is also training the deformation field though it seems? That means a optimization in 4D is executed.
> Based on my understanding of the algorithm maybe something more like:
> - 4D voxel warmup
> - Slot Instantiation
> - 4D Slot optimization
>
> Does that make sense?

---

> > ### Author Response · Authors · 2023-11-23
> >
> > Thank you for your prompt reply. While the 3D voxel warmup stage indeed involves training the deformation field within a 4D scene, we want to clarify that the additional dimension from 3D to 4D corresponds to the number of slots (𝑁) rather than the time horizon (𝑇). In other words, it goes from 3D density grids $\mathcal{F}\_{t=1}$ to object-centric 4D occupancy grids $\mathcal{V}\_{t=1}$. Hope our response can resolve your doubt.

---

### Official Review · Reviewer_B2an · 2023-11-01

**Soundness:** 3 good
**Presentation:** 3 good
**Contribution:** 2 fair
**Rating:** 6
**Confidence:** 2

**Summary:**

This paper introduces DynaVol, an attempt at unsupervised learning for object-centric representations in dynamic scenes. While most existing solutions focus on 2D image decomposition, DynaVol aims to offer a 3D perspective by leveraging object-centric voxelization within a volume rendering framework. It has a few steps to achieve this, including but not limited to canonical-space deformation optimization, and global representation learning with slo attention. The approach combines both local voxel and global features as conditions for a NeRF decoder. The proposed method achieves good results in a simulation environment and demonstrates its applicability in one real-world dataset.

**Strengths:**

1. Good attempt. The proposed DynaVol approach introduces a shift from 2D image decomposition by bringing in a 3D perspective (via NeRF). NeRF as a 3D representation beyond view synthesis is an underexplored region and I think this work makes a good effort towards this direction by doing object-centric voxelization within a volume rendering framework. Although the canonical-space deformation operation is widely seen in dynamic NeRF works, it still brings value to object-centric learning.

2. The paper is well articulated in the introduction section. I do capture the underlying motivation for the whole work, but I found a hard time understanding the two-stage training at the beginning. The authors are suggested to make the overview section and the overview figure flow better.

**Weaknesses:**

Dataset Limitations: My primary concern is the robustness of DynaVol when applied to intricate real-world scenes. The paper predominantly employs what can be described as a 'toy' dataset to validate its model. The observations made from this kind of dataset often fail to generalize to real-world applications. The mention of the HyperNeRF dataset application does lend some credence to its real-world viability. However, a more thorough exploration, perhaps using more diverse and challenging datasets (one or two demos would be enough), would provide stronger evidence of the model's capability.

Also, I wonder how important the background loss is in the pipeline. Will this method work without this loss? Will the slot decompose the background?

**Questions:**

1. Based on the results presented in Table 1, it seems that models utilizing 5 views (V=5) don't consistently surpass the performance of those with just a single view. Could the authors shed light on the underlying reason for this unexpected behavior?

2. Significance of the Warmup Stage: The paper doesn't delve deeply into the role of the warmup stage in the ablation studies. How crucial is this phase to the overall performance and efficiency of DynaVol?

3. Performance of Warmup-Stage-Only: It's intriguing to observe that the 'warmup-stage-only' variant appears to perform quite competently even though it's restricted to initial timestep observations. It's even better than other baselines reported in Table 1. Could the authors explain the factors or mechanisms behind this seemingly robust performance?

---

> ### Author Response · Authors · 2023-11-22
>
> Thank you for your valuable comments. We provide the responses below to each of the specific concerns. Please do not hesitate to let us know of any additional comments on the paper.
>
> > Q1. My primary concern is the robustness of DynaVol when applied to intricate real-world scenes.
>
> We have added new real-world experiments on the ENeRF-Outdoor dataset. Please refer to Figure 7 in Appendix.C for the qualitative results. Below, we present the quantitative comparisons in PSNR and SSIM:
>
> | Real-world scenes |       Actor1_4        |     Actor2_3      |
> | --------------------- |:-----------------------------------:|:-------------------------------:|
> | HyperNeRF  | 22.79 $\mid$ 0.759 | 22.85 $\mid$ 0.773 | 25.22 $\mid$ 0.806|
> | Ours |    **26.44** $\mid$ **0.871**      |  **26.24** $\mid$ **0.864** |
>
> > Q2. (1) How important the background entropy loss is? (2) Will the slot decompose the background?
>
> (1) On the background entropy loss:
>
> The background entropy loss is borrowed from DVGO, which aims to encourage the renderer to concentrate on either the foreground or background information. We further analyzed the impact of the background entropy loss using different loss weights (metrics: PSNR/FG-ARI):
>
>  | $\alpha_e$ |     3ObjsFall     |     6ObjsFall     |      8ObjsFall      |
>  | ---------- |:---------------------------:|:--------------------------:|:---------------------------:|
>  | 0          | 31.38 $\mid$ 94.98 |   **30.43** $\mid$ 94.25  | **30.51** $\mid$ 92.59 |
>  | 0.01 (Ours)          | **32.11** $\mid$ **96.95** |  29.98 $\mid$ **94.73**     | 29.78 $\mid$ **95.10** |
>   | 0.1          |18.20 $\mid$ 4.65  |  19.68 $\mid$ 31.63   |23.05 $\mid$ 59.01  |
>
> **Findings:** Incorporating the background entropy loss with an appropriate weight is beneficial to the scene decomposition results as it penalizes ambiguous geometric estimations. However, a higher $\alpha_e$ value can introduce instability to the training procedure.
>
> (2) Will the slot decompose the background?
>
> Yes, our proposed method is motion-aware and is capable of decomposing the background, as the background can be treated as a static component. Additionally, during the 3D-to-4D voxel expansion stage, voxel grids with significantly distinct velocities estimated by the dynamics modules tend to be initiated with different slot indices by the connected components algorithm (Please refer to Algorithm 1 in the appendix).
>
> As illustrated in Figure 3(b), our model effectively decomposes backgrounds in real-world scenarios. In the synthetic scenes in Figure 4, our model successfully decomposes the background table with an individual slot. For additional visualization results of per-slot rendering, please refer to the newly added Figure 8 in Appendix.F.
>
> > Q3. Models utilizing 5 views don't consistently surpass the performance of those with just a single view.
>
> Our motivation for using multiple views at $t=1$ stems from empirical observations in particular challenging scenarios. For instance, in 3ObjMetal, the color of a moving object with a metal surface may change rapidly across different time steps due to the drastic changes in specular reflection, which presents difficulties in learning the geometries. Incorporating additional views at the initial time step can enrich the geometric information, thereby improving the quality of the learned 3D voxel grids $\mathcal{F}\_{t=1}$ and corresponding 4D object-occupancy grids $\mathcal{V}\_{t=1}$.

---

> > ### Author Response · Authors · 2023-11-22
> >
> > > Q4. How crucial is warmup phase to the overall performance and efficiency of DynaVol?
> >
> > We introduced a new ablation study by training the entire model from scratch --- We directly performed "4D voxel optimization" with randomly initialized $\mathcal{V}\_{t=1}$ and $f\_\phi(\cdot)$. The obtained results in terms of PSNR and FG-ARI (higher values are favorable) are presented below:
> >
> > |   |         3ObjsFall          |         6ObjsFall          |         8ObjsFall          |
> > | ---------- |:--------------------------:|:--------------------------:|:--------------------------:|
> > | w/o Warmup |     31.69 $\mid$ 31.81     |     29.37 $\mid$ 20.82     |     28.55 $\mid$ 33.89     |
> > | Full model | **32.11** $\mid$ **96.95** | **29.98** $\mid$ **94.73** | **29.78** $\mid$ **95.10** |
> >
> > It is evident that excluding the warmup phase significantly impacts the final performance, especially for the scene decomposition results. We also include these comparisons in Table 4 ("4D voxel optim. from scratch") in the revised paper.
> >
> > Note: We have reorganized the entire training scheme for better clarity. Hence, the "w/o Warmup" results in the above table are obtained by excluding the first two stages below:
> > - *3D voxel warmup stage*: Corresponds to the training phase in the original warmup stage.
> > - *3D-to-4D voxel expansion stage*: Corresponds to the post-processing part of the original warmup stage (i.e., the computation of connected components).
> > - *4D voxel optimization stage*: Corresponds to the dynamic scene optimization stage in the original manuscript.
> >
> > > Q5. It's intriguing to observe that the 'warmup-stage-only' variant appears to perform quite competently even though it's restricted to initial timestep observations.
> >
> > First, it's important to clarify that, even in the warmup stage, **the model is trained with the entire image sequence** rather than being restricted to the initial time step. As shown in the updated Figure 2, it also involves an end-to-end optimization of the density grids, the bi-directional dynamics networks, and a DVGO-style renderer (more efficient than the original renderer used in D-NeRF). Therefore, considering the training data and specific model designs, it is reasonable that the "warmup-stage-only" baseline can achieve good performance.
> >
> > Additionally, it's essential to note that the "warmup-stage-only" baseline involves the "3D-to-4D voxel expansion" operations outlined in Algorithm 1 in the paper. Accordingly, the FG-AGI results validate the effectiveness of the connected component algorithm in initializing the 4D object-centric representation from the warmup of the density values in the 3D voxel grids.
> >
> > Further discussions have been incorporated in Section 4.3. For more details, please refer to our revised paper.

---

> ### Comment · Reviewer_B2an · 2023-11-23
> **Response to the rebuttals.**
>
> I thank the authors for the time they have contributed to addressing my concerns. I have increased my rating accordingly.

---

### Official Review · Reviewer_8F9K · 2023-11-01

**Soundness:** 3 good
**Presentation:** 4 excellent
**Contribution:** 3 good
**Rating:** 6
**Confidence:** 4

**Summary:**

This paper presents an optimization-based method for taking a monocular video as input and producing an object-centric 3D dynamic representation as output. The idea is to optimize a 3D voxel grid to learn the 3D occupancy/density field at each timestep, along with the warp that links each timestep to the zeroth timestep. Then, these occupancies are divided into a set of object representations using connected components, and then slot-based optimization proceeds, where  features associated with objects are averaged across the time axis, and per-timestep features are assigned to the objects via iterated slot attention. The representation is trained by rendering to the given views with a nerf-style setup, using a color loss, an entropy loss aiming to separate foreground/background, and a cycle-consistency objective on the scene flow fields. The results are impressive, both in simulation and in the two real videos.

**Strengths:**

I like this paper. It is well written, it is interesting, the method seems sensible (thought I have a few questions), and the results look good.

**Weaknesses:**

I have a variety of fairly low-level questions, which I hope the authors can address.

Overall I think the method would be easier to understand if the stages were presented in a more central way, instead of at the very end. After reading everything I am still a bit unclear on the meaning of an "episode".

Section 2 emphasizes that "we also consider scenarios where only one camera pose is available at the first timestamp". What does it mean for only one camera pose to be available? Camera poses are relative to something. A single pose might as well be the identity matrix.

The beginning of Section 3.1 says the method will be trained "without any further supervision", without defining any initial supervision. So I suppose the question is: further to what?

Section 3.1 introduces the number of objects N, as "the assumed number of objects". Is this manually chosen? Later on there is a connected components step, so I wonder if connected components could choose N instead.


It is unclear to me what exactly is happening in the warmup stage vs. the dynamics stage. From the names I might guess that the warmup stage does not involve any dynamics, but this is wrong, because it uses all timesteps and it also trains the forward and backward dynamics modules. The section also says "To obtain the dynamics information, we train an additional module f′ξ" -- but dynamics information is already obtained by f_\psi, so there is some mixup here, unless the authors do not see f_\psi as obtaining dynamics information. Finally, it's unclear how exactly the connected components and the "feature graph" connect with the rest of the method.

L_point seems like it is never defined, and only a citation is given. It would be great to describe in English what is happening there.

Section 3.3 seems to complain that prior compositional nerfs used an MLP to learn a mapping from position+direction+feature to color+density, and then proposes to do the exact same thing. Have I missed something here?

Overall I like the paper and I hope it can be cleaned up. The video results show that the segmentations are a little messy, and it seems like only 2 real videos are used, but still I think the paper is interesting and useful to build on.

**Questions:**

This is more of a personal curiosity, but: instead of learning two deformation networks (one forward, one backward) and then asking them to be cycle-consistent, would it work to learn an invertible mapping here instead? (Like with Real NVP.)

---

> ### Author Response · Authors · 2023-11-22
>
> Thank you for your comments. We hope that our responses below can adequately address your concerns regarding this paper.
>
> > Q1. The method would be easier to understand if the stages were presented in a more central way, instead of at the very end.
>
> Thank you for the suggestion. We have included more discussions on the warmup stage and dynamic scene optimization stage in Section 3.1. Please refer to the revised paper for further details.
>
> > Q2. On the meaning of an "episode".
>
> Throughout the training stage, DynaVol iteratively processes each frame in the video in chronological order and repeats: $\\{I\_1, I\_2, \ldots, I\_T\\}\_\text{episode=1}$, $\\{I\_1, I\_2, \ldots, I\_T\\}\_\text{episode=2}$, $\ldots$, where we use the term "episode" to denote one round spanning from the first frame to the last.
>
> > Q3. Section 2 emphasizes that "we also consider scenarios where only one camera pose is available at the first timestamp". What does it mean for only one camera pose to be available?
>
> The term "one camera pose" means we only have a single-view image captured by the monocular camera at $t=1$, as opposed to having multiple images from different views. We apologize for any confusion and have rephrased this statement in the revised paper.
>
> > Q4. Section 3.1 says the method will be trained "without any further supervision", without defining any initial supervision.
>
> As shown in Eq. (5), we use the pixel values of the observed video frames as the training targets for DynaVol. We here emphasize "without any further supervision" because some existing object-centric rendering approaches, such as those proposed by Stelzner et al. (2021) and Driess et al. (2022), incorporate additional information like depth maps or binary masks for each object to supervise the model.

---

> > ### Author Response · Authors · 2023-11-22
> >
> > > Q5. Section 3.1 introduces the number of objects N, as "the assumed number of objects". Is this manually chosen? Later on there is a connected components step, so I wonder if connected components could choose N instead.
> >
> > (1) The meaning of $N$:
> >
> > First, we would like to clarify that $N$ actually represents **the number of slots** rather than the number of objects (sorry for the misleading words in the paper). In all experiments, $N$ is manually pre-defined as 10 and is assumed to be larger than the number of objects in the scene.
> >
> > Using redundant slots helps alleviate the *under-segmentation* problem --- As illustrated in Figure 4: After training, the redundant slots can adaptively learn to concentrate on the background and the noises in the visual scene, and they do not contribute significantly to the final rendering results. We have clarified the meaning of $N$ in the revised paper.
> >
> > (2) Adaptatively choosing $N$ with the connected components algorithm:
> >
> > **Method:** Following the reviewer's suggestion, the computation of the adaptively determined $N$ involves the following steps:
> > - **Initial Filtering:** After the initial warmup stage, we filter out the empty voxels with low-density values.
> > - **Connected Component Identification:** We obtain $M$ connected components according to Algorithm 1 in the appendix of the revised paper. However, $M$ is much larger than the number of objects in the scene, so we have to filter out some of them.
> > - **Voxel Contribution Calculation:** For this purpose, we compute the contributions of each voxel (denoted by $m_i$) to the rendering results: $m_i =T_i \cdot \alpha_i$, where $\alpha_i=1-\text{exp}(-\sigma_i \delta_i)$ denotes the ray termination probability with volume density $\sigma_i$ and the distance $\delta_i$ between adjacent sampling points along the ray. Additionally, $T_i=\prod_{j=1}^{i-1}(1-\alpha_j)$ represents the accumulated transmittance from the Karama position to the sampling point $i$. This approach aligns with the work of "*Compressing volumetric radiance fields to 1 MB*" (Li et al., 2023).
> > - **Accumulation and Sorting:** We accumulate the $m_i$ over all voxels in each connected component and sort the accumulated contribution values of the connected components.
> > - **Obtaining Top-$N$ Components:** We filter out the connected components with lower accumulated contribution values to ensure that the sum of $m_i$ of the remaining voxels is larger than 95% of that of all non-empty voxels. This process ultimately determines the number of slots, $N$, as the number of remaining connected components.
> >
> > **Results:** Below, we compare the results of using adaptively determined $N$ with those using the manually pre-defined value of $N$. The results are presented in terms of PSNR and FG-ARI, respectively:
> >
> > | How to determine the number of slots?  |       3ObjsFall (#Objects=4)        |     6ObjsFall (#Objects=7)      |       8ObjsFall (#Objects=9)        |
> > | --------------------- |:-----------------------------------:|:-------------------------------:|:-----------------------------------:|
> > | Manually pre-defined  | **32.11** $\mid$ **96.95** ($N=10$) | **29.98** $\mid$ 94.73 ($N=10$) | **29.78** $\mid$ **95.10** ($N=10$) |
> > | Adaptively determined |     31.68 $\mid$ 95.82 ($N=5$)      | 29.85 $\mid$ **95.29** ($N=8$)  |     29.51 $\mid$ 93.16 ($N=9$)      |
> >
> > **Findings:** Our results suggest that manually pre-defining the number of slots can achieve comparable performance to the adaptive approach mentioned above while being significantly easier to implement.

---

> ### Author Response · Authors · 2023-11-22
>
> > Q6. It is unclear what is happening in the warmup stage vs. the dynamics stage. The section also says "To obtain the dynamics information, we train an additional module $f^\prime_\xi$ -- but dynamics information is already obtained by $f_\psi$, so there is some mixup here, unless the authors do not see $f_\psi$ as obtaining dynamics information.
>
> (1) Reorganizing the training stages:
>
> Yes indeed, the warmup stage also exploits the dynamics information to initialize the voxel grid features $\mathcal{V}_{t=1}$. From this perspective, we acknowledge that the names of the training stages might be somewhat misleading. We have reorganized the entire training scheme and separated it into three stages for better clarity:
> - **3D voxel warmup stage** (*the training part of the original warmup stage*): This training stage does NOT incorporate object-centric features. Instead, we train the bi-directional deformation networks $(f^\prime_\xi, f_\psi)$ and the neural renderer $N_{\phi^\prime}$ based on 3D voxel densities $\mathcal{F}_t \in \mathbb R^{N_x \times N_y \times N_z}$.
> - **3D-to-4D voxel expansion stage** (*the post-processing part of the original warmup stage*): We extend the 3D voxel grids $\mathcal{F}\_{t=1}$ to 4D voxel grids $\mathcal{V}\_{t=1}$ using the connected components algorithm. The input features for the connected components algorithm involve the forward canonical-space transitions generated by $f^\prime_\xi$ and the emitted colors by $N_{\phi^\prime}$. We give more details in Appendix.B in the revision.
> - **4D voxel optimization stage** (*the original dynamic scene optimization stage*): In this stage, we train the entire model based on a set of 4D voxel features $\mathcal{V}_t \in \mathbb R^{N \times N_x \times N_y \times N_z}$. It's important to note that the additional dimension from 3D to 4D corresponds to the number of slots ($N$) rather than the time horizon ($T$).
>
> We have made corresponding revisions in Section 3.1 (Overview of DynaVol), Section 3.4 (Descriptions of training stages), and Figure 2. Please refer to our revised paper!
>
> (2) The effect of $f^\prime_\xi$:
>
> As pointed out by the reviewer, both $f^\prime_\xi$ and $f_\psi$ are designed to capture the dynamics information, but in different time directions. We use $f^\prime_\xi$ for two purposes:
> - It allows for the computation of a cycle-consistency loss, which can enhance the coherence of the learned canonical-space transitions.
> - It provides useful input features to the subsequent connect components algorithm, representing the forward dynamics starting from the initial time step, which can improve the initialization of $\mathcal{V}_{t=1}$.

---

> ### Author Response · Authors · 2023-11-22
>
> > Q7. It's unclear how exactly the connected components and the "feature graph" connect with the rest of the method.
>
> To highlight the relationship of the computation of connected components with the rest of the method, we have reorganized the entire training scheme into three stages. As clarified in our response to Q6, the corresponding "**3D-to-4D voxel expansion stage**" acts as a bridge between the previous "3D voxel warmup stage" and the subsequent "4D voxel optimization stage". It extends the 3D density values to 4D object-centric representation:
> $$
> \text{3D-to-4D voxel expansion: } \ \mathcal{F}\_{t=1} \in \mathbb R^{N_x \times N_y \times N_z} \rightarrow \mathcal{V}_{t=1} \in \mathbb R^{N \times N_x \times N_y \times N_z}.
> $$
>
> The 3D-to-4D voxel expansion stage includes the following two steps:
> - **Feature Graph Generation:** We consider the voxel set $\\{X_k\\}\_{k=1}^{N_x \times N_y \times N_z}$ from $\mathcal{F}\_{t=1}$. By filtering out invalid locations with density values below a pre-defined threshold (inspired by DVGO), we can get the node set $\\{X_k\\}\_{k=1}^{K}$ to build the feature graph $G$. For any pair of nodes $(u,v)$ in $G$, an edge is defined between them if the following three conditions are satisfied:
>     - $u$ and $v$ correspond to spatially adjacent voxels.
>     - $u$ and $v$ result in similar emitted color $(c_u^r, c_v^r)$ produced by $N_{\phi^\prime}$ along an arbitrary ray $r$, such that $\frac{1}{\mathcal{R}}\sum\_r^\mathcal{R} d\_\text{rgb}(c_u^r, c_v^r)<\text{Threshold}\_\text{rgb}$, where $\mathcal{R}$ is the number of rays randomly sampled with different directions. We take the Euclidean distance to calculate $d\_\text{rgb}(c_u^r, c_v^r)$.
>     - $u$ and $v$ lead to similar velocities across all time steps, which can be acquired through the forward deformation network $f^\prime\_\xi$, such that $\text{max}\_t(d_\text{vel}(u,v)) < \text{Threshold}\_\text{vel}$, where we take the maximum Euclidean distance $||(f^\prime\_\xi(u,t)-f^\prime\_\xi(u,t-1))-(f^\prime\_\xi(v,t)-f^\prime\_\xi(v,t-1))||_2$ over all time steps ($t \in [2,T]$).
>
> * **Connected Components Computation:** In this step, we first feed the feature graph $G$ into the connected components algorithm to obtain $M$ clusters. We then sort the clusters by size and merge the smallest $M-N+1$ clusters. For each cluster $p$ ($1 \le p \le N$), we assign the density value of voxels in $\mathcal{F}_{t=1}$ to the corresponding channel at the same localation in $\mathcal{V}\_{t=1}$.
>
> We have included **Algorithm 1** in Appendix B to provide a detailed explanation of the 3D-to-4D voxel expansion process.
>
> > Q8. $\mathcal{L}_\text{point}$ seems like it is never defined.
>
> The formulation of $\mathcal{L}_\text{point}$ is provided in Eq. (5). Intuitively, this regularization term encourages similar color radiance values for different sampling points along the same viewing direction.
>
> Additionally, we acknowledge that it would to more reasonable to allow various sampling points on the same ray to have slightly different colors. Therefore, we tune the loss weight $\alpha_p$ and discuss of the effect of $\mathcal{L}_\text{point}$ in the appendix. As demonstrated in Table 7:
> - $\alpha_p=0.1$ (final DynaVol) outperforms $\alpha_p=0$. This suggests that $\mathcal{L}_\text{point}$ helps the training process by moderately penalizing the discrepancy among nearby sampling points on the same ray.
> - $\alpha_p=0$ (w/o $\mathcal{L}\_\text{point}$) outperforms $\alpha\_p=1$. This indicates that a strong $\mathcal{L}\_\text{point}$ may introduce bias to neural rendering and affect the final results.
>
> > Q9. Section 3.3 seems to complain that prior compositional NeRFs used an MLP to learn a mapping from position+direction+feature to color+density, and then proposes to do the exact same thing.
>
> Most prior compositional NeRFs learn a mapping function of *(Position, Direction, Feature) $\rightarrow$ (Color, Density)*, where the object-centric density values are generated by the model.
>
> In DynaVol, we slightly modify this approach by using the MLP to learn a mapping function like *(Position, Direction, Feature) $\rightarrow$ Color*, while the density values are directly retrieved from the 4D voxel grids of $\mathcal{V}^t_{\text{density}}$. The value in each grid cell indicates the occupancy probabilities of each object. This method allows for easy manipulation of objects for scene editing.

---

> > ### Author Response · Authors · 2023-11-22
> >
> > > Q10. Instead of learning two deformation networks and then asking them to be cycle-consistent, would it work to learn an invertible mapping here instead?
> >
> > Thanks for your constructive suggestion, a similar idea has been realized in the recent work OmniMotion (Wang et al., 2023). We will consider to replace the two deformation networks in DynaVol with Real-NVP (Dinh et al., 2016) in our future work.
> >
> > **References:**
> > - Stelzner et al. Decomposing 3d scenes into objects via unsupervised volume segmentation (2021).
> > - Driess et al. Learning multi-object dynamics with compositional neural radiance fields (2022).
> > - Wang et al. Tracking everything everywhere all at once (2023).
> > - Dinh et al. Density estimation using Real NVP (2016).

---

### Author Response · Authors · 2023-11-22
**General response**

We thank all reviewers for their constructive comments and have updated our paper accordingly. Please check out the new version!

Specific changes or new results include:
1. Reorganize the description of the training scheme with three stages, including the 3D voxel warmup stage, the 3D-to-4D voxel expansion stage, and the 4D voxel optimization stage (Section 3.4).
2. Refine Figure 2 to better illustrate the overall architecture and the relationship between the warmup stage to the rest of the method.
3. Clarify the meaning of $N$, which represents the number of slots instead of the number of objects (Section 3.1).
4. Include new ablation study results in Table 4 and provide further discussions on the findings (Section 4.5).
5. Include more discussions on the prior work related to DynaVol (Section 5).
6. Add Algorithm 1 in Appendix.B to present the details of 3D-to-4D voxel expansion, outlining how to initialize the 4D voxel representation using the connected component algorithm.
7. Add more real-world experiments and provide the qualitative comparisons (Figure 7 in Appendix.C).
8. Add a new figure (Figure 8 in Appendix.F) to provide further empirical analysis on the performance of slot learning.

Please do not hesitate to let us know of any additional comments on the paper.

---

### Author Response · Authors · 2023-11-23

Dear reviewers,

Thank you once more for your valuable reviews. We have posted our clarification and response. Please kindly let us know if our response and revised paper have addressed your concerns. We are more willing to answer your remaining concerns and questions.

We would appreciate it if you could consider increasing your rating if you find our responses helpful.

Best,
Authors

---

### Meta-Review · Area_Chair_1UU1 · 2023-12-05

**Metareview:**

This paper introduces DynaVol, a method for object-centric 4D (i.e. 3D+dynamics) scene representation learning from monocular video.

This is a very timely & important yet challenging topic, and the reviewers all agree that the paper makes progress in this area. Reviewer 8F9K in particular highlights that the results are impressive, both on synthetic as well as on real-world data. I further agree with the reviewers that the paper is generally well written and of high quality.

Several concerns were raised by the reviewers regarding the method as well as the experimental validation:

The primary concern by reviewer B2an, namely the applicability of DynaVol to more complex real-world scenes, was in my view successfully addressed by the authors, who added additional convincing real-world results during the rebuttal phase.

Reviewer FQjp remarks that the final method is rather complex and that the model description in the initial submission was not very clear. The authors have addressed this aspect as well, and reviewer FQjp agrees that the updated version is clearer.

The only reviewer who recommended rejection (R p1WX) raised several concerns around 1) ablation studies, 2) clarity of novelty, and 3) clarity of the introduction (regarding importance of the problem). While the reviewer failed to engage in the rebuttal/discussion process, I believe that all their concerns were successfully addressed in the rebuttal. The other reviewers further agree that this is indeed an important problem.

Overall, I think the paper meets the bar for acceptance. It will be a valuable addition to the literature on 3D scene representation learning for dynamic scenes.

**Justification For Why Not Higher Score:**

While the paper makes solid progress on this particular problem, it does not necessarily constitute a breakthrough.

**Justification For Why Not Lower Score:**

The paper makes solid progress on an important/challenging problem of interest to the ICLR community. It is well-written and well-executed. Most concerns raised by the reviewers in their initial assessments were successfully addressed during the rebuttal.

---

### Decision · Program_Chairs · 2024-01-16

Accept (poster)